# ON-OFF receptive fields in auditory cortex diverge during development and contribute to directional sweep selectivity

Joseph Sollini [1], Gaëlle A. Chapuis[1], Claudia Clopath [1] & Paul Chadderton [1,2]

Neurons in the auditory cortex exhibit distinct frequency tuning to the onset and offset of sounds, but the cause and significance of ON and OFF receptive field (RF) organisation are not understood. Here we demonstrate that distinct ON and OFF frequency tuning is largely absent in immature mouse auditory cortex and is thus a consequence of cortical development. Simulations using a novel implementation of a standard Hebbian plasticity model show that the natural alternation of sound onset and offset is sufficient for the formation of non-overlapping adjacent ON and OFF RFs in cortical neurons. Our model predicts that ON/OFF RF arrangement contributes towards direction selectivity to frequency-modulated tone sweeps, which we confirm by neuronal recordings. These data reveal that a simple and universally accepted learning rule can explain the organisation of ON and OFF RFs and direction selectivity in the developing auditory cortex.

[1] Department of Bioengineering, Imperial College London, London SW7 2AZ, United Kingdom. [2] School of Physiology, Pharmacology and Neuroscience, Biomedical Sciences Building, University of Bristol, University Walk, Bristol BS8 1TD, United Kingdom. These authors contributed equally: Joseph Sollini, Gaëlle A. Chapuis. These authors jointly supervised this work: Claudia Clopath, Paul Chadderton. Correspondence and requests for materials should be addressed to C.C. (email: c.clopath@imperial.ac.uk) or to P.C. (email: p.chadderton@bristol.ac.uk)

Appearances and disappearances are both salient events for sensory processing[1–4], and neurons in many sensory systems exhibit robust responses to stimulus initiation and termination[5–11]. In the mature primary auditory cortex (A1), neurons respond to both the onset and offset of sound via activation of non-overlapping populations of synapses[5]. These synaptic inputs have discrete frequency tuning, ensuring that individual A1 neurons exhibit distinct frequency selectivity to sound onset and offset[5–7,12]. A1 is topographically organised, and the frequency selectivity of individual neurons is largely determined by their positions within this tonotopic gradient[13–15]. Given fixed positions within this map, it is surprising that neurons exhibit different ON and OFF selectivity, and the mechanistic basis and functional significance of this organisation are not understood.

Individual neuron ON and OFF RFs are typically adjacent but non-overlapping with respect to frequency, indicating a high degree of specificity in ON/OFF organisation. This specificity may have an anatomical substrate; for instance, ON and OFF inputs could arise from discrete, but adjacent, regions of the auditory thalamus. In this scenario, differences in ON and OFF tuning should be present at the onset of hearing (postnatal days 13–17[16];). Alternatively, pruning of synaptic inputs or changes in synaptic strength later in development could drive RF reorganisation. Here, we combine electrophysiological recordings and computational modelling to reveal the cortical mechanisms underlying the organisation of ON and OFF RFs. We demonstrate that discrete ON/OFF frequency tuning develops following cortical exposure to sound, and complementary Hebbian plasticity of ON and OFF inputs is sufficient for RF reorganisation. Further, we show that the maturation of ON and OFF receptive fields organisation contributes to the functional selectivity of individual neurons to higher-order stimulus features. Specifically, ON/OFF RF arrangement is related to directional selectivity for slow, ethologically relevant frequency modulations even when synaptic inhibition is reduced, potentially providing a novel mechanism for cortical encoding of vocalisations[17].

## Results

**Developmental divergence of A1 receptive fields**. In order to compare ON and OFF RFs in A1 neurons from developing and adult animals, extracellular population recordings were made from young (P15–23; 1–10 days after hearing onset and when hair cell activity has matured[16]; $N = 7$) and adult (>P60; $N = 10$) mice. Action potentials evoked by the onset and offset of pure tones (Fig. 1a, b) were used to construct frequency response areas (FRAs) and determine characteristic frequencies (CF) for individual neurons. Evoked responses to sound offsets were observed in approximately equal fractions in young and adult mice (65 ± 7%, $n = 642$, and 58 ± 8%, $n = 258$, of neurons in young and adult mice, respectively; see Methods). In individual neurons from young mice, ON and OFF FRAs were qualitatively similar (Fig. 1c), and CFs for onset and offset were commonly in agreement (Fig. 1e). In contrast, A1 neurons in adult mice exhibited segregated but adjacently aligned ON and OFF FRAs[5–7] (Fig. 1d, e). We measured ON/OFF segregation by calculating absolute differences in octaves (oct) between ON and OFF CFs for individual neurons ($CF_{diff}$). Distributions of $CF_{diff}$ were statistically different between young and adult neurons (Fig. 1f; $CF_{diff}$: Kolmogorov–Smirnov test, $D = 0.148$, $p = 0.0017$), with significantly smaller values in the young population ($CF_{diff}$: Wilcoxon signed rank test, $Z = 4.03$, $p = 5.5 \times 10^{-5}$). ON/OFF CFs most commonly showed direct correspondence in the young population (young $CF_{diff}$ mode = 0 oct), whereas ON/OFF CFs were distributed adjacently in mature cortical neurons (adult

$CF_{diff}$ mode = 0.25 oct). CF disparity was organised with respect to ON CF, neurons with low ON CFs were more likely to have higher frequency OFF CFs and vice versa (Supplementary Figure 1). This relationship between ON CF and $CF_{diff}$ was significantly correlated in both adults and juveniles (PPMC, adult: $r = 0.6136$, $p = 1.46 \times 10^{-77}$, juvenile: $r = 0.3022$, $p = 3.04 \times 10^{-6}$). Developmental differences between ON/OFF CF were reflected in the overall arrangement of ON and OFF FRAs in young and adult animals, such that FRA overlap was significantly reduced in adult ($\mu = 41\%$) vs. young ($\mu = 57\%$) mice (Fig. 1g, Mann–Whitney $U$ test = 8.36, $p = 6.4 \times 10^{-17}$). Taken together, these results demonstrate reorganisation of the relative tuning of A1 neurons to sound onset and offset during development, ON and OFF RFs show a high degree of similarity at the onset of hearing but diverge following cortical exposure to sound.

**Hebbian plasticity accounts for RF developmental divergence**. How do ON and OFF RFs become segregated during development? We hypothesised that functional reorganisation could occur either via selective pruning of synapses (i.e. anatomical reorganisation), or by modifying the strengths of ON and OFF inputs. Because the selectivity of cortical neurons can be altered by activity-dependent plasticity[18–20], we explored whether Hebbian learning underlies the shift in ON/OFF RF arrangement. To test this idea, we designed a computational model consisting of a rate-based feedforward network where neurons receive a series of excitatory and inhibitory ON and OFF inputs[21] across a range of frequency channels (Fig. 2a). In this model, both excitatory and inhibitory synaptic efficacies are plastic according to Hebbian learning; when pre- and postsynaptic neurons fire together, their weights increase, otherwise they decrease[20–24] (Fig. 2b; see Methods). To reproduce conditions prior to hearing onset, neurons in the network were initially driven by synchronised spontaneous (i.e. not sensory-evoked) activity from ON and OFF inputs, leading to the development of input selectivity. At this stage (i.e. at hearing onset), ON and OFF RFs were matched (Fig. 2c). We then tested whether sound-driven inputs permit the refinement of ON and OFF RFs. To mimic postsynaptic integration in vivo, the temporal profile of onset- and offset-evoked activity was modelled to match the time course of sensory-evoked excitatory postsynaptic potentials (EPSPs) in A1[25]. Within this scheme, summation of onset- and offset-evoked activity can occur when offsets rapidly follow onsets for transient sounds, and when onsets rapidly follow offsets during brief gaps. In nature, acoustic environments are characterised by ongoing initiations and terminations of power within specific frequency channels[4,26,27], so the network was driven with sound sequences that switched on and off randomly at given frequencies (Fig. 2a). When presented in our model, sequences composed of fluctuating onsets and offsets drove plasticity of excitatory and inhibitory ON and OFF inputs, leading to receptive field reorganisation (Fig. 2d, e; Supplementary Figure 2). Sound presentation produced progressive adjustments in synaptic weights and gradual divergence of ON and OFF CF (Fig. 2f), in a manner that was robust to changes in the underlying model parameters (Supplementary Figure 3). The same pattern of reorganisation was observed if sound-evoked input was evoked in one frequency channel at a time (single channel condition) or across all channels (overlapping sound condition; Supplementary Figure 3a; also see Methods), corresponding to single or multiple overlapping sound sources, respectively. We compared ON/OFF frequency selectivity at early and late time points in the simulation (time point: $t = 1500$, 'Young', and $t = 100000$, 'Adult') and found increased ON/OFF CFs differences in mature networks (Fig. 2g, h;

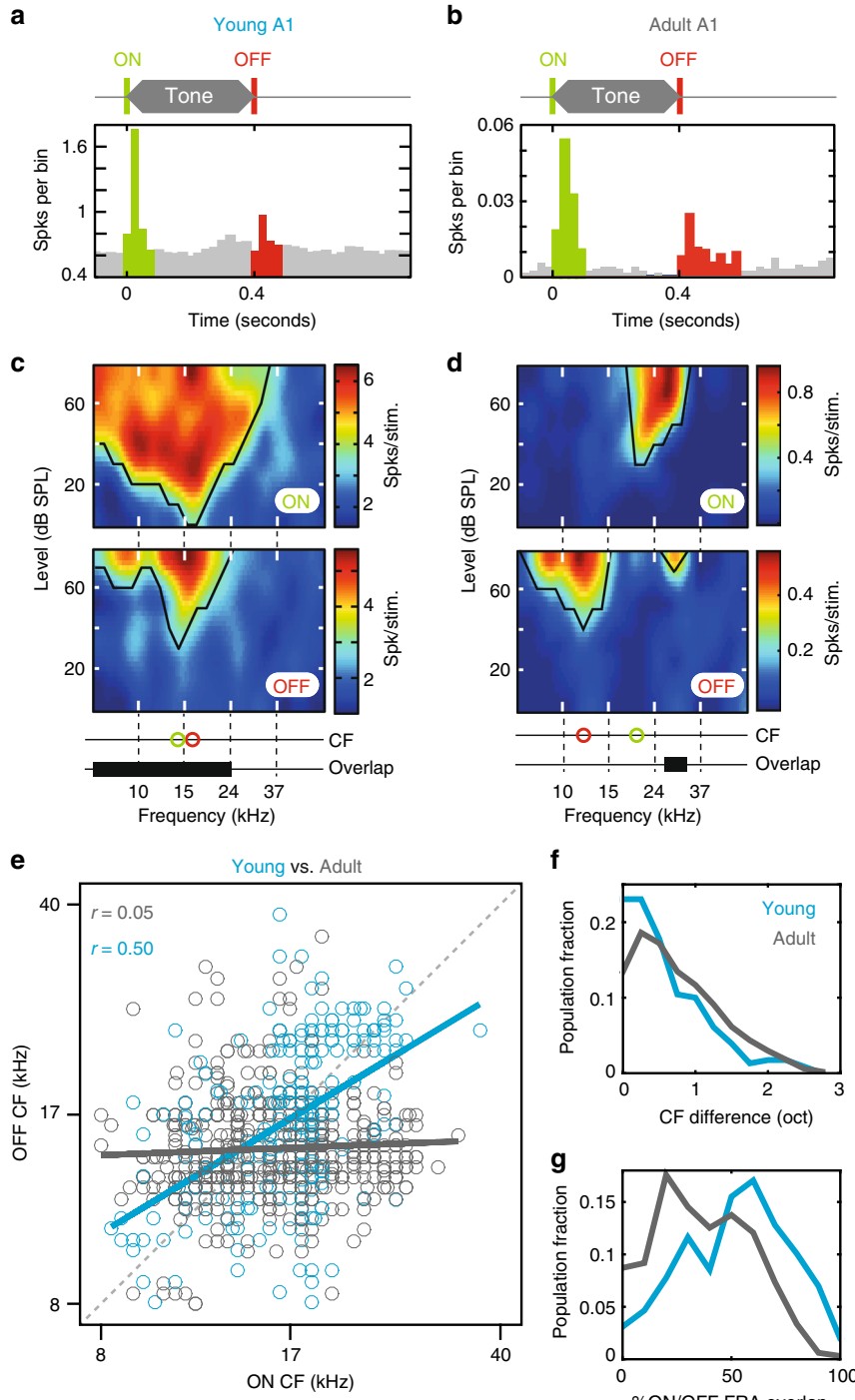

**Fig. 1** Developmental divergence of ON and OFF receptive fields in primary auditory cortex. **a**, **b** Example peri-stimulus time histograms of spiking activity in A1 neurons of young (P15–23; **a**) and adult (>P60; **b**) mice responding to pure-tone onset (green) and offset (red). **c** Single neuron frequency response areas (FRAs) for sound onsets (top) and offsets (middle) in young mouse. Bottom, characteristic frequency (CF) was similar for onset and offset (green and red circles), and FRAs were overlapping (black bar). **d** Single neuron FRAs for sound onsets (top) and offsets (middle) in adult mouse. Bottom, CFs were segregated for onset and offset (green and red circles), and FRA overlap was limited (black bar). **e** Relationship between ON and OFF CF for individual A1 neurons in young (blue) and adult (grey) mice. The dashed diagonal line represents a perfect correspondence between ON and OFF CF. **f** Distribution of the absolute difference between ON and OFF CF for neurons in young (blue) and adult (grey) mice. **g** FRA overlap in young and adult mice. Receptive field boundaries are largely overlapping early in development (blue) and later become segregated (grey)

Kolmogorov–Smirnov test, $p = 0.0082$), as experimentally observed (Fig. 1f).

We next explored which aspects of the model were necessary to reproduce the biological features of ON/OFF RF divergence. Importantly, as with the experimental data, simulated ON and OFF RFs remained adjacently positioned even though they were segregated (Fig. 2h). We considered that sequential activation of ON and OFF inputs in an alternating pattern was necessary for this property of RF reorganisation. To test this, we used an alternative stimulation paradigm in which model neurons were

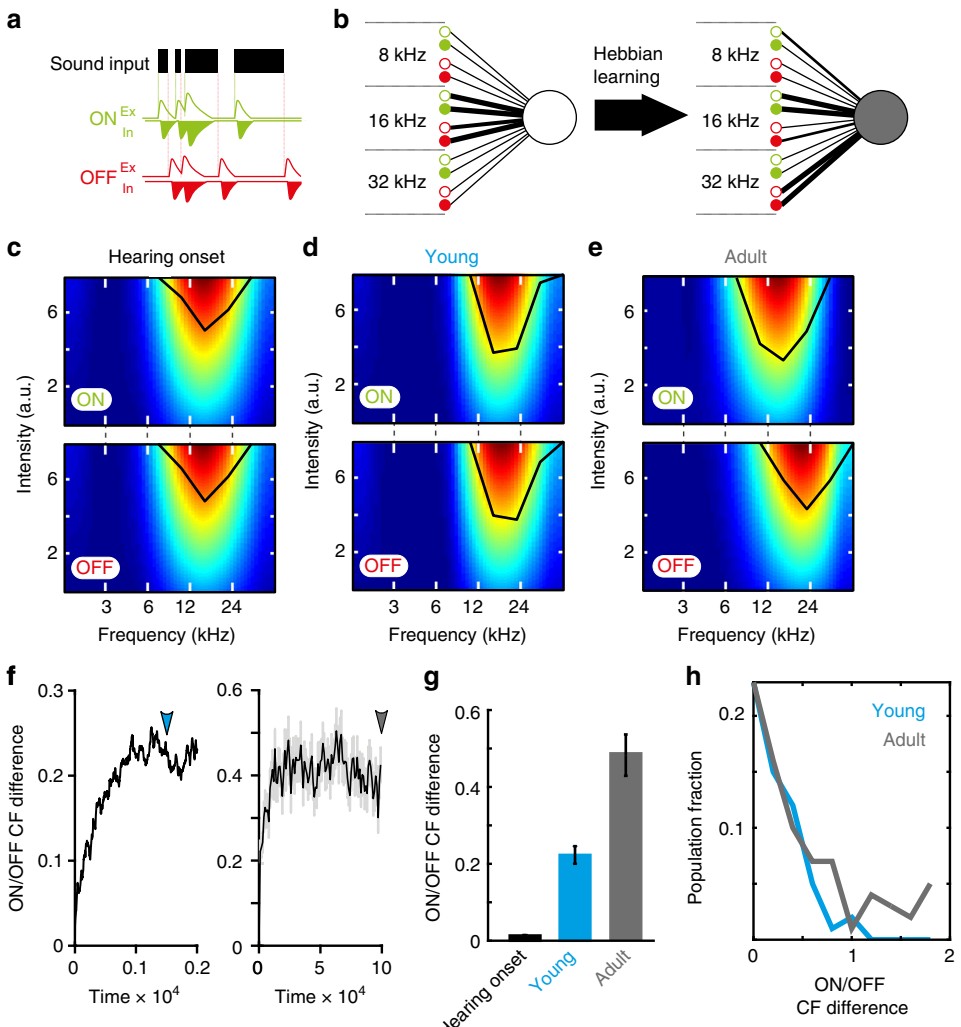

**Fig. 2** Hebbian plasticity can account for developmental divergence of ON and OFF RFs. **a** Model schematic of synaptic input to investigate influence of sound exposure on ON/OFF RFs. Top, Neurons receive sound-evoked synaptic activity, driven by a sound input sequence (black) that randomly switches ON (green dashed lines) and OFF (red dashed lines) in separate frequency channels. Middle: Excitatory (Ex; upward) and inhibitory (In; downward) ON inputs are evoked by sound onsets. Bottom, Excitatory (Ex; upward) and inhibitory (In; downward) OFF inputs evoked by sound offsets. **b** Simulated network is comprised of a neuron (white; left) that receives 10 excitatory and 10 inhibitory ON inputs (green, open and filled circles, respectively), and 10 excitatory and 10 inhibitory OFF inputs (red, open and filled circles, respectively). Only three ON and OFF input channels are shown for clarity. Synaptic weights undergo Hebbian learning during sound presentation. At simulation end (adult condition in grey, right), ON and OFF synaptic weights have diverged. **c** Example ON (top) and OFF (bottom) FRAs at the start of the simulation ('Hearing Onset'). FRAs are identical. **d** Example ON (top) and OFF (bottom) FRAs at an early stage ($t = 1500$) of the simulation ('Young'). **e** Example ON (top) and OFF (bottom) FRAs at the end of the stimulation ($t = 100,000$; 'Adult'). **f** Temporal evolution of RF divergence in single channel condition. Left, ON/OFF CF divergence during early stage of simulation (steps 0–2000). Blue arrow: $t = 1500$ ('Young' time point shown in **d**). Right, complete evolution of ON/OFF CF divergence. Grey arrow: $t = 100000$ ('Adult' time point shown in **e**). **g** Mean absolute difference between ON and OFF CF at Hearing Onset (black), Young (blue) and Adult (grey) time points ($n = 100$) in single channel condition. **h** Distribution of absolute ON/OFF CF differences in Young and Adult neurons ($n = 100$) following sound presentation in single channel condition

presented with one channel sequence for ON inputs ('Independent ON channel') and a different sequence for OFF inputs ('Independent OFF channel'; Fig. 3a; see Methods). In this non-ethological scenario (i.e. where sound onsets and offsets need not necessarily alternate), tuning curves again underwent reorganisation but now ON/OFF RFs were no longer arranged adjacently (Fig. 3b; Kolmogorov–Smirnov test, $p = 9 \times 10^{-9}$). These results indicated that naturally occurring sequences of onset and offset drive RF divergence in a manner that maintains the close apposition of ON and OFF RFs. We next assessed the contribution of evoked synaptic inhibition to RF divergence by removing inhibitory ON and OFF inputs from our model (Fig. 3c). Under this scenario, ON/OFF RF divergence still

occurred (Fig. 3d; Young vs. Adult time point: K–S test, $p = 0.047$) indicating that evoked synaptic inhibition is not necessary for developmental RF reorganisation. Finally, we tested whether the Hebbian learning rule itself was necessary for RF reorganisation. To do this, we modelled RF divergence using an alternative plasticity rule—homoeostatic synaptic scaling[28] (see Methods)—without Hebbian learning. Sound-evoked input in this condition failed to produce the divergence of ON/OFF RFs (Supplementary Figure 4; Hearing Onset vs. Adult time point: K–S test, $p = 0.96$). These experiments demonstrate that a combination of sound-driven activity and Hebbian learning is a plausible mechanism underlying the maturation of receptive fields in A1 in the absence of anatomical reorganisation.

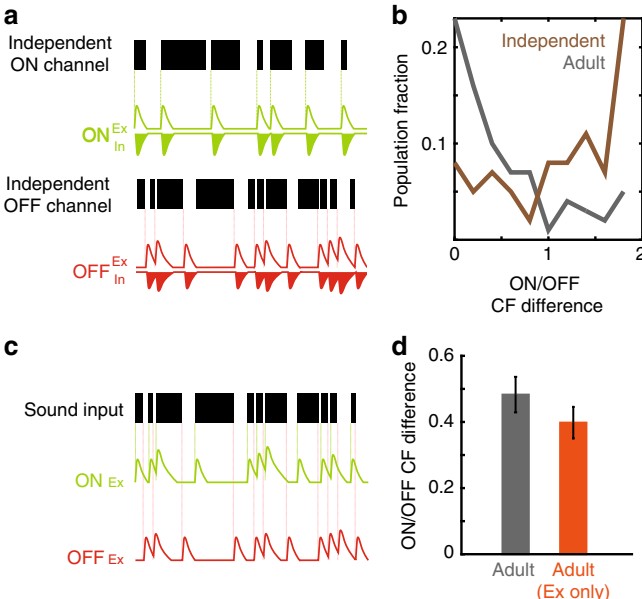

**Fig. 3** Determinants of ON/OFF RF divergence in silico. **a** Input scheme to investigate role of sound alternation in RF divergence. In contrast to Fig. 2 (trained with input sequences where ON and OFF always followed each other), the network is presented with independent ON and OFF sequences where ON/OFF inputs need not necessarily alternate. **b** Final distribution of ON/OFF CF differences for neurons ($n = 100$) trained with independent ON/OFF input sequences (brown). Adult population trained with natural alternating inputs shown for comparison (grey). **c** Input scheme to investigate role of fast evoked synaptic inhibition in RF divergence. Neurons receive evoked synaptic activity, driven by a sound sequence (black) that randomly switches ON (green dashed lines) and OFF (red dashed lines). Excitatory ON (green) and OFF inputs are evoked by sound onsets. Inhibitory inputs are omitted from the model. **d** Final mean difference between ON and OFF CF ($n = 100$ cells) in excitation-only model (Ex only, orange). Adult excitation-inhibition results shown for comparison (grey)

**ON/OFF RF arrangement and FM sweep direction selectivity.** What is the functional significance of ON/OFF RF segregation in mature animals? We hypothesised that the arrangement of ON and OFF RFs could influence how cortical neurons respond to frequency-modulated (FM) sounds[29] (Fig. 4a), which are etho-logically relevant in vocalisation[30]. FM sounds will activate ON inputs as they enter a preferred frequency channel and activate OFF inputs when they exit. Our rationale was that the relative arrangement of ON and OFF RFs could increase directional selectivity to FM sweeps; if a neuron is selective to higher frequency onsets than offsets (ON higher than OFF), then ascending tone sweeps will exit the preferred-OFF channel and enter the preferred-ON channel coincidently. Conversely, if a neuron is selective to lower frequency onsets than offsets (ON lower than OFF), then descending tone sweeps will exit the preferred-OFF channel and enter the preferred-ON channel coincidently. In both cases, the close temporal association of OFF and ON activation could lead to summation of ON/OFF inputs and thus generate enhanced firing. We tested this prediction by presenting ascending (UP) and descending (DOWN) tone sweeps to our model neurons. Neurons that had relatively lower frequency ON selectivity (ON lower than OFF) had significantly higher firing rates to descending sweeps than ascending sweeps (Fig. 4b). Conversely, neurons that had relatively higher frequency ON selectivity (OFF lower than ON) responded most strongly to ascending sweeps (Fig. 4c). Overall, our simulations predicted a clear relationship between the alignment of ON/OFF frequency

selectivity and FM direction sensitivity (Ex-In model: $r = -0.63$; $p = 2.3 \times 10^{-12}$; Fig. 4d–f), and this relationship was robust to changes in the underlying model parameters (Supplementary Figure 5). As with developmental RF divergence, this property was preserved when evoked synaptic inhibition was removed from the model (Ex only model: $r = -0.85$; $p = 7.0 \times 10^{-29}$), suggesting that the tuning of excitatory ON and OFF synaptic inputs are sufficient to account for FM direction selectivity.

We tested whether the predictions of our simulation were borne out in the in vivo characteristics of A1 neurons. We measured evoked responses in young and adult A1 to ascending and descending FM sweeps at different velocities (range: ± 2.2–70 octaves/second) and compared directional sensitivity to ON/OFF RF arrangement in neurons that possessed V-shaped ON and OFF RFs (young: $n = 20$ cells in $N = 3$ mice; adult: $n = 30$ cells in $N = 4$ mice; see Methods). Notably, in the adult population, the difference between ON and OFF preferred frequency was a good predictor of direction selectivity. Fourteen out of the thirty cells had lower frequency ON vs. OFF (ON lower than OFF), and ten out of these fourteen cells preferred downward FM sweeps (Fig. 5a; Supplementary Figure 6a). Sixteen of the thirty cells had higher frequency ON vs. OFF (OFF lower than ON), and all sixteen cells preferred upward FM sweeps (Fig. 5b, Supplementary Figure 6b). Direction selectivity in these neurons was only prominent at slower speeds (2.2–17.5 oct/sec; Fig. 5c) where remarkably the magnitude of direction selectivity (assessed via direction selectivity index; DSI) was correlated with the relative distribution of ON/OFF preferred frequencies ($r = -0.61$ at 2.2 oct/sec; $p = 0.0004$; Fig. 5d). In the young population, we observed some directional FM selectivity with a bias towards upward FM sweeps[31], but the alignment of ON/OFF RFs was not predictive of this selectivity.

We confirmed the contribution of ON/OFF RF alignment to direction selectivity in adult, but not young, A1 using multi-variable linear regression models to test for the relative contribution of different functional neuronal properties in predicting direction selectivity index (DSI). The tested properties included the strength and variability of evoked responses to sound onset and offset, the bandwidth and overlap of ON and OFF RFs, and the octave difference between ON and OFF CF (see Methods). The importance of each property for DSI prediction was assessed via the absolute value of their normalised coefficients, and the proportional reduction of error generated when adding the property as predictor. The performance of the model in predicting the DSI was assessed via the adjusted R-square, which considers the number of parameters used. For the population of adult A1 neurons, ON/OFF CF difference had the largest coefficient and accounted for a large proportion of the error reduction in DSI prediction (Supplementary Table 1). By eliminating functional properties that degraded model perfor-mance, we generated a restricted model (Supplementary Table 2). The functional properties included in the restricted model comprised the magnitude of the ON-evoked response, the bandwidth of the OFF RF, %ON/OFF RF overlap, and ON/OFF CF difference. As seen in the full model, the normalised coefficient of ON/OFF CF difference was the largest. Moreover, the other properties included in the model were poor in predicting the DSI when used in single parameter model (adjusted R-square < 0.05). These results indicated that the performance of the restricted model mainly relied on the linear relationship between the ON-OFF CF difference and the DSI, confirming that this parameter was the most important linear predictor of the DSI in adult mice. When the same procedures were performed for the population of young A1 neurons, no such relationship was found. Instead, %ON/OFF RF overlap was the most important predictor, however the linear relationship

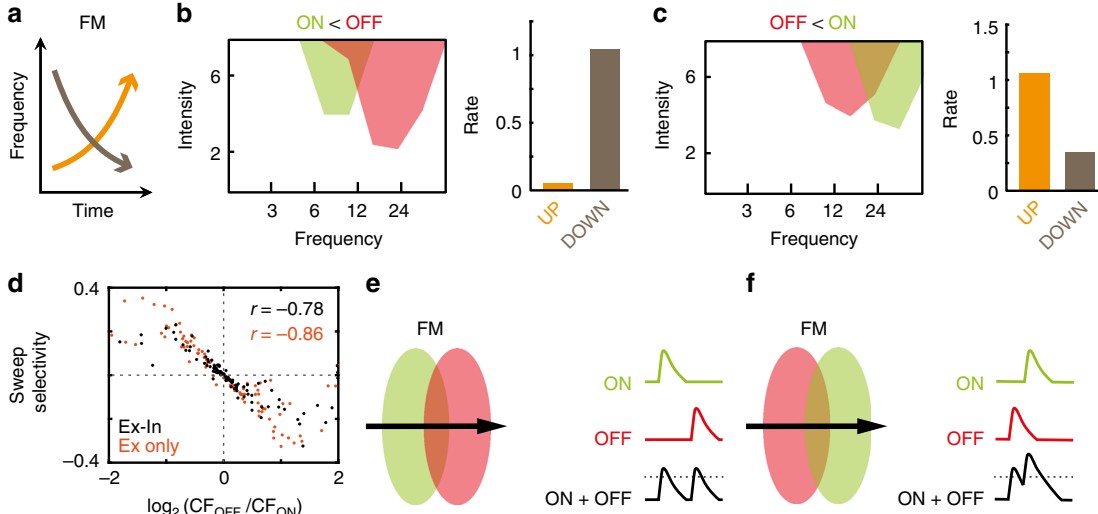

**Fig. 4** Proposed functional significance of segregated ON and OFF RFs in A1. **a** Schematic of frequency-modulated (FM) sweep inputs. Modelled adult neurons are presented with either UP (orange) or DOWN (brown) FM sweeps. **b** Example model neuron's FRA outline (left), where ON CF is at a lower frequency than OFF CF. Mean firing rate is greater for sweeps going DOWN rather than UP (right). **c** Same as **b**, but for example neuron where ON CF is at a higher frequency than OFF CF. In this case, the neuron prefers sweeps going UP. **d** Sweep selectivity as a function of ON/OFF CF difference for 100 cells in network with excitatory and inhibitory inputs (as in Fig. 2a; black dots), and excitatory-only network (as in Fig. 3c; orange dots). **e-f** Schematic of mechanism underlying sweep selectivity. **e** Example ON/OFF RF arrangement where ON RF (green ellipse) is selective for lower frequencies than OFF RF (red ellipse). When a FM sweep is going UP (black arrow), ON (green) and OFF (red) inputs are activated at different times. The sum of the inputs (black) is only just above the neuronal threshold (dashed black line). **f** Example ON/OFF RF arrangement where ON RF is selective for higher frequencies than OFF RF; when a FM sweep is going UP, OFF and ON inputs are activated in rapid succession and summate far above neuronal threshold

between this property and the DSI was poor (adjusted R-square < 0.06; see Supplementary Table 3). Together these results support the relationship between ON/OFF RF organisation and FM direction selectivity in adult A1, as predicted in silico.

**Cortical inhibition not required for direction selectivity.** Rapid cortical inhibition has previously been implicated in the expression of FM direction selectivity[32]. However, our model suggested that excitatory ON/OFF RF arrangement alone could contribute to FM direction selectivity. We tested this prediction by targeting one class of inhibitory interneurons, parvalbumin-positive interneurons (PV+), using a pharmacogenetic silencing approach. PV + neurons are fast spiking and preferentially target perisomatic regions of excitatory pyramidal cells, providing strong feedforward synaptic inhibition[33]. In auditory cortex, PV+ neurons are well-tuned for frequency and exhibit short response-latencies[34], leading to the possibility that evoked PV+ inhibition can 'out run' and/or overwhelm synaptic excitation evoked by FM sweeps in non-preferred directions[32,35] to enforce direction selectivity. We tested this proposal in A1 by selectively expressing the inhibitory DREADD receptor, hM4i, in PV+ cells. AAV-DIO-hM4i-mCherry virus was unilaterally injected into the auditory cortex of *Pvalb-Cre* mice to target the hM4i receptor to A1 PV + cells (Fig. 6a). After allowing sufficient time for transduction and expression (eight weeks, see Methods), we performed electrophysiological recordings to measure evoked responses in A1 to pure tones and FM sweeps. Evoked activity was compared between an initial recording period (Baseline), and after intraperitoneal injection of the DREADD agonist, clozapine-N-oxide (CNO; 5 mg/kg; see Methods), when the excitability of PV+ neurons was reduced. Following CNO injection, the magnitude of evoked local field potentials (LFPs) increased in hM4i-PV+ but not sham control animals (Fig. 6b–d—normalised ON response at BF post-CNO: $2.24 \pm 0.14$ in hM4i-PV+ animals vs. $1.10 \pm 0.02$ in sham control animals; Wilcoxon signed rank test: $p = 2.2 \times 10^{-6}$; $n = 56$ and $n = 24$, respectively), consistent with reduced cortical

inhibition in hm4i-PV+ animals only. However, the overall tuning profile of LFP FRAs were unchanged (Fig. 6e; non-significant change in ON CF between Baseline and post-CNO: $0.04 \pm 0.03$ oct; Wilcoxon signed rank test: $p = 0.26$). Multiunit (MU) recordings revealed that CNO injection produced a robust and significant increase in spontaneous and evoked firing rates in hM4i-PV+ but not control animals (Fig. 6f, g; normalised spontaneous firing rate post-CNO: $3.14 \pm 0.27$ in hM4i-PV animals vs. $1.13 \pm 0.08$ in sham control animals, $p = 2.0 \times 10^{-7}$; normalised evoked ON firing rate post-CNO: $2.48 \pm 0.26$ in hM4i-PV animals vs. $1.17 \pm 0.09$ in sham control animals, $p = 2.5 \times 10^{-4}$; normalised evoked OFF firing rate post-CNO: $2.81 \pm 0.29$ in hM4i-PV animals vs. $1.13 \pm 0.11$ in sham control animals, $p = 5.3 \times 10^{-7}$; Wilcoxon signed rank tests; $N = 7$ and $N = 3$, respectively). The overall structure of MU FRAs were also unchanged (Fig. 6h, non-significant change in evoked ON CF: $0.09 \pm 0.05$ oct, $p = 0.41$), indicating that PV+ inhibition plays a prominent role in regulating excitability but not sensory tuning of local A1 neurons[34,36]. We further found no significant differences in ON/OFF RF bandwidth and best frequency threshold level for single units recorded pre- and post- CNO injection ($p > 0.1$ for all variables tested, Wilcoxon rank sum test; $n = 30$ and 29 units pre- and post-injection). Having confirmed the fidelity of the pharmacogenetic perturbation in hM4i-PV+ animals, we assessed directional sensitivity in single units following CNO injection and compared to ON/OFF RF arrangement in neurons that possessed V-shaped ON and OFF RFs ($n = 29$ cells in $N = 7$ mice; see Methods). Within this population, neurons exhibited direction selectivity, and the alignment of ON/OFF RFs in individual neurons remained a predictor of direction selectivity (Fig. 6i, Supplementary Figure 6c). Overall, the magnitude of direction selectivity (assessed via direction selectivity index; DSI) was correlated with the relative distribution of ON/OFF preferred frequencies ($r = -0.40$ at 2.2 oct/sec; $p = 0.03$; Fig. 6j). We therefore conclude that the segregated arrangement of ON/OFF

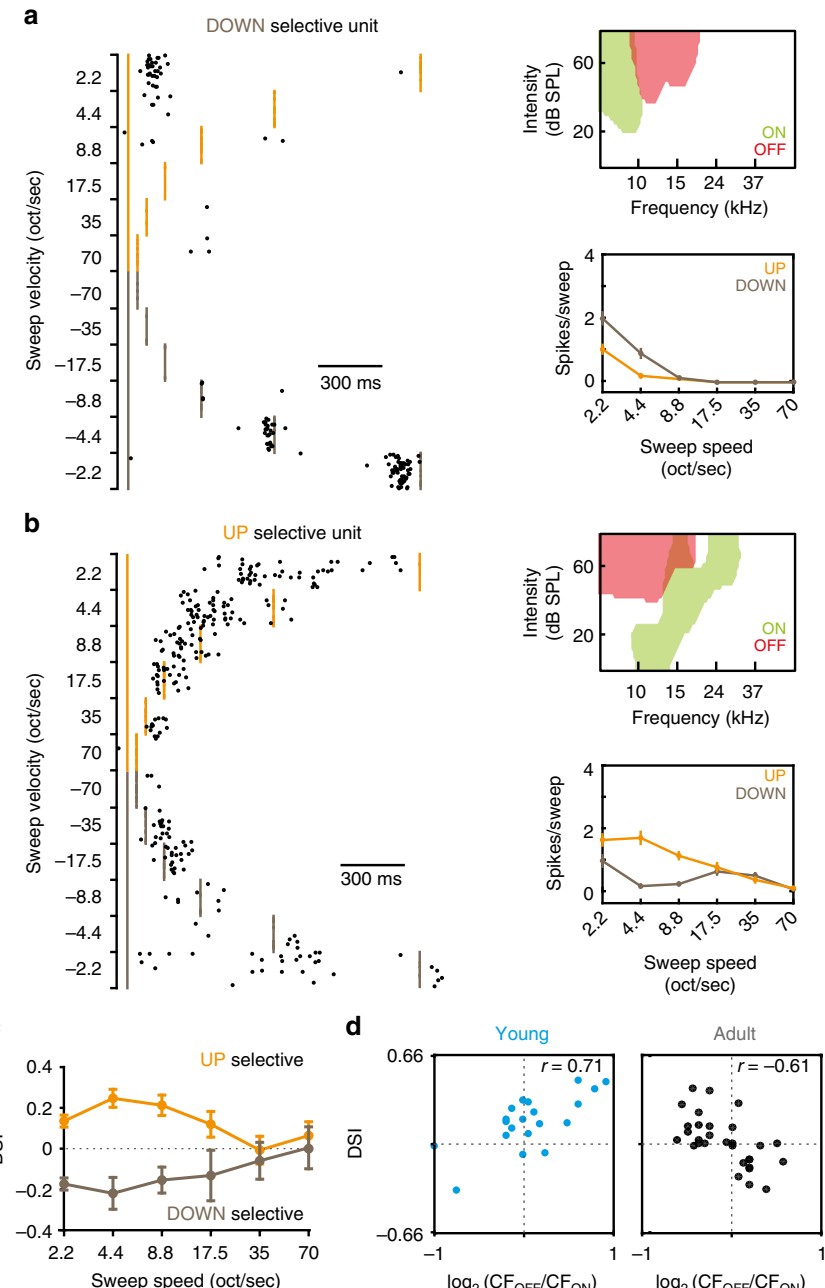

**Fig. 5** Arrangement of ON and OFF receptive fields confers direction selectivity to slow frequency-modulated sweeps in vivo. **a** Response of a DOWN-selective unit to 3-octave FM sweeps. Left, raster plot of spiking responses to different velocities. Orange and brown ticks indicate beginning and end of UP and DOWN sweeps, respectively. Top right, outline ON and OFF FRAs for the same unit. Bottom right, mean spiking responses to sweeps at different speeds for UP and DOWN directions. **b** Same as **a**, but for an UP-selective unit. **c** Direction selectivity index (DSI) at different speeds for the UP-selective units (orange) and DOWN-selective units (brown) recorded in adult A1. Positive DSI indicates selectivity for UP FM sweeps. Only units with measureable ON and OFF receptive fields were included. In these units, direction selectivity was observed at slower speeds only (>35 oct/sec). **d** Relationship between directional selectivity and ON/OFF receptive field organisation in young (left) and adult (right) A1 neurons. Octave spacing between ON and OFF CF is indicated on the bottom axis. Positive values indicate that OFF CF is higher than ON. Each point represents an individual unit (adult: $n = 30$; young: $n = 20$). DSI values represented are calculated at the lowest speed (2.2 oct/sec). In adult A1, directional selectivity and ON/OFF receptive field arrangement correspond as predicted in silico

RFs in A1 neurons can confer directional sensitivity to FM sounds, as predicted by our theoretical model.

## Discussion
In the auditory cortex, sound onsets and offsets activate distinct populations of synapses that have different frequency tuning[5].

Here we show that ON and OFF tuning in individual neurons overlaps at hearing onset and diverges following exposure to sound. Our simulations suggest that Hebbian plasticity can fully account for developmental changes in the organisation of ON and OFF RFs. Functional reorganisation via changes in input strength is an efficient mechanism as it does not require discrete and specific anatomical reorganisations of ON and OFF projections.

Such reorganisation also provides flexibility for further experience-dependent learning in auditory cortex[18–20,37]. These results exemplify that precise temporal patterning of sensory events can influence the expression of experience-dependent plasticity and provide an essential cue in shaping the functional organisation of cortical circuitry.

How does the natural alternation of sound onset and offset lead to RF divergence? Onsets and offsets activate discrete populations of synapses[5]—this means that even if ON inputs and OFF inputs share identical tuning properties at hearing onset, these input classes can independently undergo plasticity and have the potential to diverge from one another. Natural sounds have both beginnings and ends and, therefore, activate ON and OFF inputs in turn. In the model, this means that ON and OFF inputs for a given frequency channel are activated sequentially and approximately equally. As a result, potentiation (and depression) of ON and OFF inputs also occurs sequentially and approximately equally, limiting RF divergence. However, in our model, instances

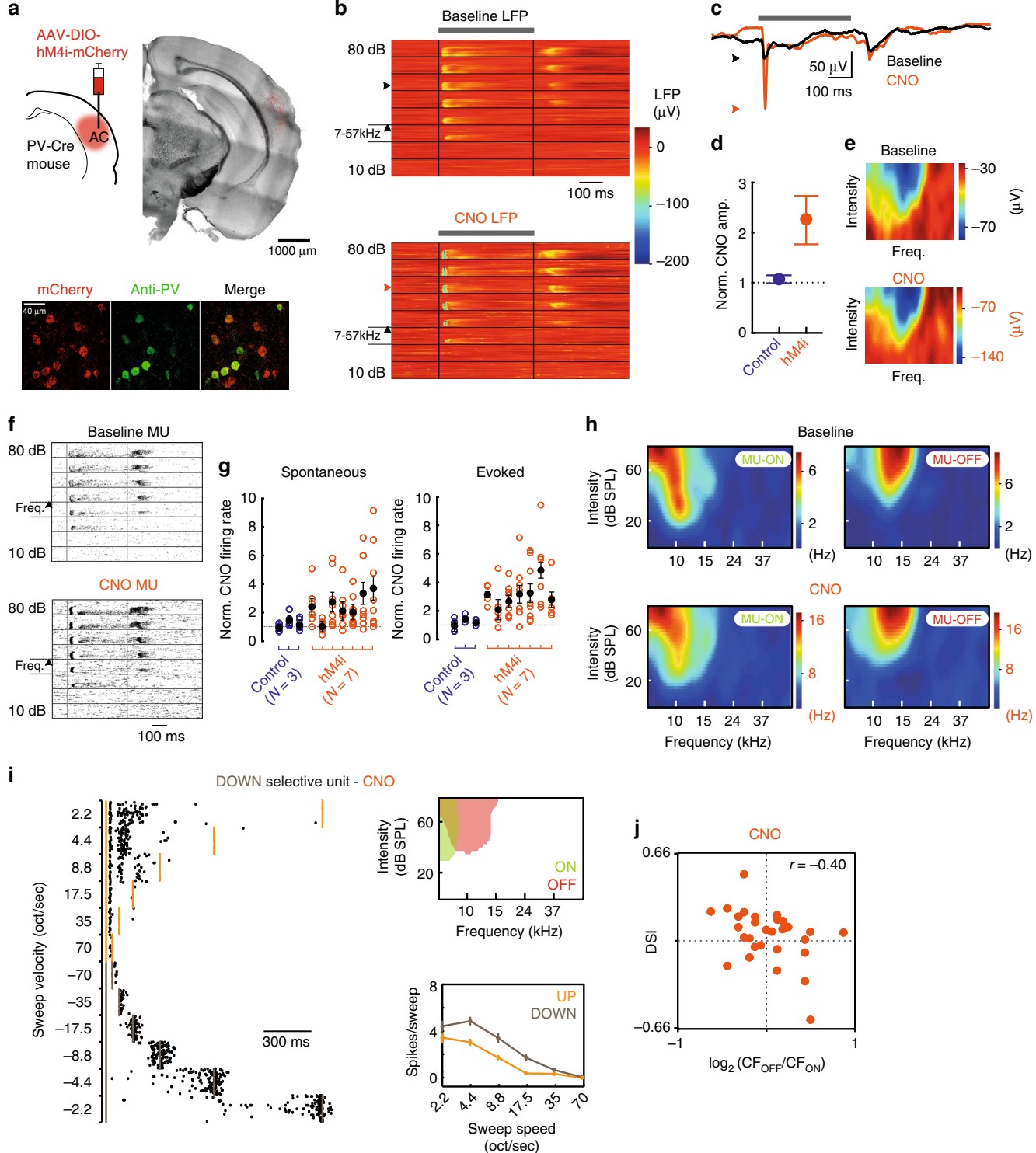

occur when onsets or offsets evoke only subthreshold activity and are, therefore, not subject to synaptic potentiation via Hebbian learning. In these instances, the strength of ON and OFF inputs diverge slightly, allowing RFs to gradually segregate while remaining closely apposed (Fig. 2f–h). Hebbian plasticity takes place slowly over many stimulus presentations[22] leading to slow gradual RF divergence. In the version of the model where ON and OFF inputs are treated as truly independent channels (an unrealistic scenario in nature; Fig. 3a), ON and OFF inputs are not required to follow one another, removing the constraints on RF divergence and enabling more drastic rearrangements to occur (Fig. 3b). Therefore, it is the inherent temporal order of onsets and offsets (i.e. one must follow another) that is sufficient to drive RFs to segregate but remain aligned in the frequency domain. In an ethological setting, we propose that exposure to alternating sound onsets and offsets during early days and weeks of life lead to gradual divergence of ON and OFF RFs.

Processing of sound onset and offset by the auditory system is particularly important for perception[2–4]. The segregated arrangement of ON and OFF RFs within individual neurons allows these two different signals to be encoded independently by the same A1 population. However, our results also indicate that interactions between ON and OFF inputs contribute to the selectivity of single neurons to higher-order features. Frequency modulation is a common feature of natural sounds, and is particularly relevant in vocalisation, including speech. Several mechanisms have been put forward to account for cortical FM directional selectivity, the most prominent of which require the involvement of cortical inhibition[29,32]. Direction selectivity in A1 is influenced by mismatched frequency tuning of synaptic inhibition (measured as asymmetrical 'inhibitory sidebands' surrounding excitatory regions of the FRA;[32,38]), but has also been observed when GABAergic inhibition is pharmacologically blocked[38]. Here we used pharmacogenetics to suppress inhibition provided by fast spiking cortical interneurons. Our experiments reveal that CNO injection in control animals does not alter neural activity in A1 (Fig. 6d, g), ruling out non-specific effects of metabolically converted CNO (i.e. clozapine acting upon other receptors in the brain) influencing these results[39]. In hM4i-PV mice only, CNO increased neural excitability in A1, consistent with suppression of cortical inhibition. In these animals, our results show that specifically reducing the excitability of fast spiking PV+ interneurons does not abolish direction selectivity in the auditory cortex (Fig. 6i, j). Our results do not rule out that alternative sources of cortical inhibition, such as somatostatin-expressing interneurons[40,41], may play a direct role in the expression of FM direction selectivity. However, we suggest that while inhibitory sidebands may play an essential role in the formation of directional selectivity at subcortical processing stations[35], and at rapid speeds[32], this mechanism may not be applicable at slower speeds in the auditory cortex[31]. The majority of neurons in mouse A1 show preferences for slower speeds of FM[42–44], which is a common characteristic of mouse vocalisations[30]. Interestingly, all neurons with measureable ON and OFF RFs displayed directional selectivity at slow but not fast speeds in this study (Fig. 5c). Therefore, the apposition of ON and OFF RFs during the functional maturation of A1[37] may enforce direction selectivity to slow FM sounds in a manner that is consistent with our model predictions. Such a contribution for ON/OFF RFs in auditory cortical direction selectivity may be analogous to their role underlying orientation selectivity in primary visual cortex[8,9].

This is the first theoretical and experimental study to demonstrate how plasticity may shape neuronal selectivity to two different and separate classes of input. We reveal that ON and OFF inputs interact with one another and despite identical starting points, RFs reorganise to become segregated but adjacent. Our results highlight that to understand the selectivity to higher-order stimulus features in A1, it is necessary to consider both ON and OFF RFs. Furthermore, because all neurons in the neocortex show selectivity to more than one stimulus feature, our results suggest that it is necessary to consider the rich tuning properties of individual cells to fully understand neuronal selectivity.

## Methods

**In vivo electrophysiology.** The care and experimental manipulation of animals was performed in accordance with institutional and United Kingdom Home Office guidelines. Twenty adult (P60-90; 13 C57BL/6 and 7 *Pvalb*-IRES-Cre) and seven juvenile (P15–23; C57BL/6) mice of both genders were used in this study. Mice were anaesthetised with fentanyl/midazolam/medetomidine mixture (0.05, 5.0 and 0.5 mg/kg) and were fixed in place via a headplate secured to the cranial surface using adhesive (Histoacryl; Braun Corporation, USA). A small craniotomy was made directly above auditory cortex (adult: centred 2.7 mm posterior of Bregma; juvenile: scaled from adult assuming adult Bregma-Lambda distance of 3.7 mm). The dura was removed and silicon microelectrodes comprising multiple tetrodes (A32, 4 × 2Tet; NeuroNexus, USA) were advanced via micromanipulator (IVM, Scientifica, UK) into auditory cortex at an angle perpendicular to the cortical surface. The mean depth of recording in adults was 568 ± 47 μm (mean ± SD; $N =$ 14). In juvenile animals, recordings were performed when spikes and evoked LFP deflections were visible on the bottom four sets of tetrodes. Data were acquired via Digital Lynx 16SX system (Neuralynx, USA) and stored on a PC. Blinding and randomisation of neurophysiological data was not performed in this study.

**Auditory stimulus presentation.** Auditory stimuli were pre-generated and calibrated (5–100 kHz flat spectrum ± 1.5 dB SPL) using Matlab (Mathworks, USA) and presented free-field (ES1; Tucker Davis Technologies, USA) via an RZ6 Processor (using RPvdsEX software; Tucker Davis Technologies, USA). Pure-tone

**Fig. 6** Direction selectivity and relationship with ON/OFF RF arrangement are maintained following pharmacogenetic inhibition of parvalbumin-positive interneurons. **a** Top, AAV1/2-hSyn-DIO-hM4Gi-mCherry injection into auditory cortex of *Pvalb*-IRES-Cre mice. Bottom, co-expression (merged image, right) of mCherry (red, left) and PV-antibody (green, middle). **b** Tone-evoked local field potential (LFP) recording in A1 at baseline (black) and following clozapine-*N*-oxide (CNO; orange) injection. Grey bar indicates period of tone presentation. Black vertical lines indicate onset and offset. Responses are grouped by intensity (10–80 dB SPL) and ordered by frequency (7–57 kHz). Arrows indicate intensity/frequency combination shown in **c**. **c** Tone-evoked LFP at baseline (black) and post-CNO (orange) for intensity/frequency combination indicated in **b** (average of five repeats). Arrows indicate peak ON-evoked LFP amplitude in each condition. **d** Normalised change in ON-evoked LFP amplitude at BF following CNO injection for recordings from sham- (Control, blue; $N = 3$) and AAV-DIO-hM4i- (hM4i, orange; $N = 7$) injected mice. **e** ON FRA at baseline (top) and post-CNO (bottom) for LFP recording shown in **b,c**. RF structure is maintained but note overall increase in response magnitude post-CNO (calibration bar, right). **f** Tone-evoked multiunit (MU) activity from a single tetrode at baseline (top) and post-CNO (bottom). Black vertical lines indicate tone onset and offset. Responses are grouped by intensity (10–80 dB SPL) and ordered by frequency (7–57 kHz). **g** Normalised change in spontaneous (left) and evoked ON (right) MU firing rate post-CNO in sham- (Control, blue) and AAV-DIO-hM4i- (hM4i, orange) injected mice. Coloured open circles correspond to individual tetrodes (eight per recording). Filled black circles correspond to individual animal means (Control, $N = 3$; hM4i, $N = 7$). **h** ON and OFF FRAs at baseline (top) and post-CNO (bottom) for MU recording shown in **f** RF structure is maintained despite overall increase in firing rate following CNO injection. **i** Response of DOWN-selective unit to FM sweeps post-CNO injection. Left, spiking responses to UP and DOWN sweeps. Right, outline FRAs mean spiking responses to sweeps for same unit. **j** Directional selectivity is correlated with ON/OFF RF organisation following pharmacogenetic inhibition of PV+ interneurons. DSI values at 2.2 oct/sec

FRAs were measured using 25 different frequencies (400 ms duration, 7–56 kHz, with 0.125 octave spacing), eight different sound levels (10–80 dB SPL at 10 dB steps), with a 1 s inter-stimulus interval. Stimulus order was randomly selected with each stimulus repeated 6–15 times in standard-, and five times in each pharmacogenetic recording session. Direction selectivity was measured in four C57BL/6 and seven *Pvalb*-IRES-Cre mice using frequency-modulated (FM) sweeps spanning three octaves, varying in velocity (12 different velocities: ± 2.2–70 octave/second). FM sweeps were created using the *chirp* Matlab function, and were presented at 60 dB SPL. Positive velocity represents an UP sweep (from 7 to 56 kHz), and a negative velocity represents a DOWN sweep (from 56 to 7 kHz). The order of the stimuli was randomly selected with each stimulus repeated 30 times.

**Pharmacogenetics.** Homozygous *Pvalb*-IRES-Cre mice (JAX stock #008069, $N = 7$) of both genders were used. At eight weeks of age, the auditory cortex was unilaterally injected with AAV1/2-hSyn-DIO-hM4Gi-mCherry to selectively express the inhibitory Designer Receptor Exclusively Activated by Designer Drug (DREADD) in parvalbumin-positive (PV+) interneurons. Mice were anaesthetised with 1–2% isoflurane under aseptic conditions and held using ear bars on a stereotaxic frame (Angle 2, Leica Microsystems, Germany). A small craniotomy was performed −2.60 mm posterior and 4.30 mm lateral of Bregma. A glass pipette was lowered vertically in auditory cortex and 500 nL of virus was injected in the auditory cortex at two injection sites (injection rate of 50 nL/min). The pipette was initially inserted to a depth of 1010 um and 250 nL of virus was injected; 5 min later, the pipette was retracted to a depth of 800 um where the remaining 250 nL of virus was injected. The pipette was then removed, and the tissue sutured. Analgesia was administered via intraperitoneal injection (Carprofen, 5 mg/kg). The animal was left to recover, and the virus left to express for eight weeks. The recording paradigm in pharmacogenetic experiments consisted of a 25 min baseline recording where auditory stimuli were presented, followed by intraperitoneal injection of the DREADD agonist Clozapine-N-Oxide (CNO; 5 mg/kg body weight). Following a silent 20 min period post- injection, auditory stimuli were again presented for 25 min. Thereafter, the probe was advanced deeper in the brain (typically 100 μm), and another 25 min recording was performed. For sham control experiments, C57/BL6 adult mice were used ($N = 3$) in lieu of transduced PV-Cre animals.

**Histology.** Mice were deeply anaesthetised with Euthatal and transcardially perfused with PBS (0.1 M) followed by 4% paraformaldehyde solution (PFA, wt/wt in PBS). Brains were extracted, placed in PFA solution and stored at 4 °C overnight. Coronal brain sections (60–100 μm) were cut using a vibrating microtome (VT1000 S, Leica Microsystems, Germany). Immunochemistry on mouse brain section was performed using rabbit anti-Parvalbumin antibody (Swant, code no. PV 27) and Alexa Fluor 488 goat anti-rabbit IgG antibody (Life Technologies, code no. A11008). Fluorescence images were acquired on a Leica SP5 laser-scanning confocal microscope. Multi-tile scanned images were performed using a ×10 air objective and ×40 oil objective. No zoom was used. The laser power and gain of the PMTs were adjusted to avoid photobleaching and image saturation.

**Data analysis.** SpikeDetekt (http://sourceforge.net/projects/spikedetekt/) and KlustaKwik were used to detect and sort single units. Clusters were manually inspected using Klusters and reclustered when necessary. Clusters that contained > 1% of spikes within a 1 ms interspike interval were rejected (http://neurosuite.sourceforge.net). FRAs were visually assessed and categorised manually (selected in isolation, blind and at random). Those that did not contain clear structure for both onset and offset and those that were considered noisy were rejected. Raw FRAs were smoothed (3 × 3 pyramidal window) and iso-response curves (FRA edges) were classed as a 30% change from baseline firing rate[45]. CF was taken as the frequency yielding a defined response at the lowest signal level and bandwidths were defined as the width of the defined region 30 dB above threshold. Best frequency (BF) was measured as the frequency/level that yielded the highest spike count.

DSI was defined as $(r1 − r2)/(r1 + r2)$[32], where r1 is the mean number of spikes triggered by the UP sweep at a given FM speed, and r2 the mean number of spikes triggered by the DOWN sweep at the same speed. The time window for measuring spiking responses started at sound onset and ended 100 ms after sound offset. Neurons were considered direction selective if the absolute value of the mean DSIs at the two lowest FM speeds (2.2 and 4.4 octave/second) was above 0.05.

Data are presented as mean ± standard error of the mean unless otherwise stated.

**Modelling of experimental data.** A linear model was generated based on the equation $y \sim = a_1x_1 + a_2x_2 + \ldots + a_nx_n$, where $y$ represented the variable to predict (i.e. DSI), $x_j$ represented the different predictor properties, and $a_j$ represented the estimated coefficients corresponding to each of these properties. Initially 10 functional neuronal properties were used, namely (1) the firing rate increase evoked by onset of a pure tone of $CF_{ON}$ frequency at 60 dB, (2) the firing rate increase evoked by offset of a pure tone of $CF_{OFF}$ frequency at 60 dB, (3, 4) Fano factor of each of these two responses, (5) spontaneous firing rate and (6) associated Fano factor, (7, 8) the bandwidths of ON and OFF RFs measured at 30 dB, (9) the percentage overlap between ON and OFF RFs, and (10) the octave difference

between ON and OFF CFs. The number of observations used to estimate the model coefficients was equivalent to the number of cells with ON and OFF RFs, i.e. 30 for adult mice and 20 for young mice. To measure the individual contribution of a property $x_j$ towards the model performance, the component $x_j$ was removed from the original equation. The truncated equation $y_{trunc} \sim = a_1x_1 + \ldots + a_{j-1}x_{j-1} + a_{j+1}x_{j+1} \ldots + a_nx_n$ was then used to predict DSI values, from which the residual sum of squares $SSE_{truncated}$ were computed. The proportional reduction of error was defined as $(SSE_{truncated} — SSE_{full})/SSE_{full}$, where $SSE_{full}$ is the residual sum of squares of the model including all predictors. To retrieve normalised coefficients $a_j$, the data of each property were z-scored.

**Analysis of pharmacogenetic response.** The LFP and MU comparative analysis in *Pvalb*-IRES-Cre and control animals were computed using only the recordings where baseline and post-CNO injection traces were acquired at the same penetration depth. Mean changes in evoked LFP and MU responses were calculated per tetrode, using only the responses to stimuli contained within the FRA edges of the baseline condition. The mean change for one tetrode was defined as $mean(R_2/R_1)$, where $R_1$ is a vector containing the mean responses obtained for each stimulus selected in the baseline period, and $R_2$ the mean responses obtained in the post-CNO period. A value above 1 therefore denotes an increase in activity post-CNO injection compared to baseline. A LFP response was quantified as the global minimum over the response window (0–300 ms after sound onset for evoked ON responses). A MU response was quantified as the number of spikes present in the response window (0–100 ms after sound onset/offset for evoked ON/OFF responses). Mean change in baseline MU activity for one tetrode was defined as $S_2/S_1$, where $S_1$ and $S_2$ are the mean number of spikes detected in the 100 ms window preceding the pure tones presented in the baseline and post-CNO periods, respectively. Tetrodes with less than 1 spike on average in the baseline ($S_1 < 1$) were excluded from the comparative analysis. Changes in LFP and MU CF are computed as $log_2(CF_2/CF_1)$, i.e. the octave difference between the CF calculated in the baseline condition ($CF_1$) and the CF calculated post-CNO injection ($CF_2$). Due to increased baseline firing activity after CNO injection (including presumably spiking activity from previously silent cells), single unit clustering was performed separately post-CNO and well-isolated single units were included in analysis of post-CNO frequency tuning and direction selectivity.

**Neuron model and network.** We simulate threshold-linear neurons with a rate-based description of neural activity. Each neuron receives 40 inputs $x_i$, consisting of $N = 10$ excitatory ON inputs ($x_i^{ON^e}$), 10 excitatory OFF inputs ($x_i^{OFF^e}$), 10 inhibitory ON inputs ($x_i^{ON^i}$) and 10 inhibitory OFF inputs ($x_i^{OFF^i}$). The voltage $u$ is computed by the weighted sum of its inputs $x_i$,

$$u = \sum_{i=1}^{10} w_i^{ON^e} x_i^{ON^e} + w_i^{OFF^e} x_i^{OFF^e} + w_i^{ON^i} x_i^{ON^i} + w_i^{OFF^i} x_i^{OFF^i},$$ where $w_i^{ON/OFF^{i/e}}$ denote the ON/OFF excitatory/inhibitory weights (or synaptic efficacies), respectively. The output of the neuron $y$ is the voltage thresholded, i.e. $y = u − \theta$, if $u > \theta$; $y = 0$ otherwise, where the threshold is set to $\theta = 2.5$.

To model the fact that neighbouring inputs in frequency space are correlated, we generate the inputs $x_i$ assuming that they each have a similar tuning to stimuli. These stimuli are modelled as 10 time-dependent activities $s_j(t)$ (which corresponds to a sound amplitude at a given frequency, $i$). The activity of input $i$ is calculated by a sum of the stimulus channels, weighted with tuning strengths $x_i^{ON/OFF^{i/e}}(t) = \sum_j T_{ij} s_j^{ON/OFF^{i/e}}(t)$. The input tuning is Gaussian: $T_{ij} = e^{-\frac{(i-j)^2}{2\sigma}}$ for $i$ and $j$ going from 1 to 10. The parameter $\sigma = 1.5$ denotes the tuning width. To avoid boundary effects, we have a circular boundary condition of the 10 $ON^{i/e}$ inputs and of the 10 $OFF^{i/e}$ inputs, meaning that input 1 and input 10 are neighbours.

**Hebbian learning rule.** For excitatory synapses, we model a standard Hebbian learning rule, $\Delta w_i^e = \alpha^e x_i^e y + \eta w_i^e$, where $\alpha^e = 10^{-4}$ is the excitatory learning rate of the Hebbian term consisting of the presynaptic activity $x_i^e$ times the postsynaptic activity $y$. The second term models synaptic noise or turnover[46,47]. The parameter $\eta$ is a random variable chosen at every time step from a uniform distribution between −0.0025 and 0.0025. Because Hebbian plasticity is inherently unstable[48], this rule has to be complemented by a weight-limiting mechanism. We therefore bound the weights between 0 and 1, and the sum of the $ON^e$ weights and $OFF^e$ weights are kept constant (i.e. L1 norm of 2[21,49]), similar to synaptic scaling[28]. The weights are initialised uniformly to $\frac{1}{2N}$.

For the inhibitory synapses, we use the rate-based version of the rule of Vogels et al.[24], confirmed experimentally in auditory cortex[20], $\Delta w_i^i = \alpha^i x_i^i (y − \rho)$, where $\alpha^i = 10^{-5}$ is the inhibitory learning rate, $\rho = 0.01$ is a constant. This rule has been shown to develop excitatory/inhibitory co-tuning[21,24]. As for the excitatory synapses, the ON/OFF inhibitory synapses undergo a L1 normalisation, here of −1, and the synapses are constrained to be negative. Finally, the weights are initialised at zero.

To test an alternative plasticity mechanism, we simulate the network with synaptic scaling (Supplementary Figure 4). At every time step, we scale down multiplicatively all the excitatory weights by a factor $1–10^{-5}$ if the output $y$ is bigger than the average output doing the whole simulation $<y>$ or up multiplicatively by a factor $1 + 10^{-5}$ if the output $y$ is below $<y>$.

**Inputs before hearing onset**. Before sound-evoked activity, neurons $s_i(t)$ receive spontaneous activity. These input patterns are generated from filtered uniform noise between $-0.5$ and $0.5$ with a time constant of 5, by subtracting a constant $\theta_x = 0.1$, setting all negative values to zero and then rescaling the signal to have a mean firing rate of 1 (arbitrary units). ON and OFF spontaneous activity is the same for each frequency band and for excitation and inhibition. The model was simulated for 100,000 time steps.

**Sound-evoked inputs**. The sound is generated so that it can either be ON or OFF for one given frequency $s_i$. At each time step, if the sound is ON, the sound can be switched OFF with a probability $p_{ON} = 1/50$. Similarly, if the sound is OFF, there is a probability $p_{OFF} = 1/50$ of switching it ON. When the sound switches ON at a given frequency, the corresponding ON input increases of 1, and decays exponentially otherwise, with a decay time constant of $\tau = 10$. Similarly, when the sound switches OFF, the corresponding OFF input increases of 1, and decays otherwise with the same time constant $\tau$. Once the inputs have been generated, they are rescaled to have a maximum value of 40. Excitatory and inhibitory inputs are the same. The network was simulated for 100,000 time steps. We tested two types of sound inputs which yielded similar results: one case where only one frequency channel can be ON at the same time (main paper) and another case where each frequency channel can turn ON and OFF independently ($p_{OFF} = 1/500$), so that multiple frequencies are played at the same time (Supplementary Figure 3a and Supplementary Figure 4).

**Analysis of modelled data**. We simulated 100 cells. To plot FRAs, constant intensities $s_i$ are played from 19.5 to 65. The difference between ON and OFF is computed as the centre of mass of the ON FRA minus the centre of mass of the OFF FRA. As the indices $i = 1,..,10$ corresponds to 5 octaves, an average difference of 2 corresponds of 1 octave. To test sweep selectivity, the mean firing rate of the neuron $y$ is measured while a sweep is going UP or DOWN for 3000 time steps. The sound is presented for one time step in a frequency band $i$, then 50 ms later in the next frequency band ($i+1$ for sweep going UP and $i-1$ for sweep going DOWN, note the circular boundary condition) etc. ON and OFF inputs $x$ are calculated as above and maximal values are rescaled to 2.

**Data availability**. All neurophysiological data are available from the authors. The model is available on ModelDB.

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

## Acknowledgements

We thank Bill Wisden for provision of the DREADD virus; Adel Haddad, Altina Fazliu and Yu Liu for tissue processing and microscopy; and Susu Chen, Troy Margrie and Sam Wass for feedback on the manuscript. This work was supported from the following sources: Bioengineering departmental PhD studentship to G.C.; Wellcome Trust Investigator Award (200790/Z/16/Z) and grants from the Leverhulme Trust and Biotechnology and Biological Science Research Council (BB/N013956/1 and BB/N019008/1) to C.C.; Medical Research Council Career Development Award (G1000512) and grants from the Human Frontier Science Program and the Biotechnology and Biological Science Research Council (BB/N008871/1) to P.C.

## Author contributions

J.S., G.C., C.C. and P.C. conceived and designed the study. J.S. and G.C. performed electrophysiological experiments. J.S., G.C., C.C. and P.C. conducted data analysis and interpretation. C.C. built the model and performed the simulations. P.C. wrote the manuscript with contributions from J.S., G.C. and C.C..

## Additional information

**Competing interests:** The authors declare no competing interests.

