## [Peer Review File · Nature Communications]

Editorial Note: Reviewer #3 was added after the second round of review with expertise in both computational modeling and auditory physiology.

Reviewers' comments:

Reviewer #1 (Remarks to the Author):

Here Sollini et al. examine development of the mouse auditory cortex; specifically, the emergence of ON and OFF responses to pure tones. With extracellular recordings in anesthetised animals they show that in young mice (P17-23), ON and OFF fields are more overlapping compared to adult frequency-response areas. A fairly small-scale model based on Hebbian plasticity is proposed to account for the development of ON/OFF responses, and the authors examine the relation between ON/OFF responses and FM sweep direction selectivity.

There are some interesting data here, and relating different receptive field properties, examining them over development, and trying to replicate their emergence in a modeling framework are all noble and important efforts. However, I think that this study at present is underdeveloped, suffering from a lack of connecting qualities- specifically, the authors make no attempt to test their model, which seems a bit simplistic and leaves out important aspects of processing for receptive fields (namely, inhibitory responses).

Critiques:

1) No effort is made to test the model experimentally. If Hebbian plasticity can fully account for the reorganization of ON/OFF receptive fields, then presenting similar sound patterns as used in the model to drive Hebbian learning (Fig. 2c) to juvenile animals, should result in more rapid divergence of ON/OFF fields in these animals. Can the authors experimentally demonstrate this? There are also predictions for spiking patterns evoked by these sounds, the potential engagement of NMDA receptors or other aspects important plasticity, but none of these issues are examined here.

2) ON/OFF responses have different excitatory-inhibitory relations that might shape these responses and/or depend on the maturation of inhibitory inputs, but the model takes into account only excitatory inputs. In general there's little exploration of the parameter space or important qualities of the model that might inform or constrain experiments. Inhibition might also be important for FM sweep selectivity.

3) The baseline activity in the recording shown from the juvenile animal is much higher compared to that shown from an adult (Fig. 1a,b). Is this representative? Increased baseline activity in juvenile animals could potentially account for the ON/OFF fields overlap by imposing high-pass filtering of sensory-evoked responses. Is there a way to show that Hebbian plasticity, and not the decrease in baseline activity, is indeed responsible for the divergence of ON/OFF receptive fields in adult animals?

4) Stereotaxic coordinates and recording depth should be included in Methods. It will be helpful to include histological verification of the recording sites in the results.

5) The 'n' is a bit low here; the critical data come from a total of four juvenile animals.

Reviewer #2 (Remarks to the Author):

In "Hebbian plasticity underlies developmental divergence of ON-OFF receptive fields and direction selectivity in auditory cortex", Sollini et al examine the development and functional significance of ON-OFF responses in mice. Using extracellular recordings in anesthetized juvenile and adult animals, they propose that the frequency shift between ON-OFF responses arises from hearing experience, and that

a general Hebbian learning model can account for this shift. Furthermore, they propose that frequency shifts in ON-OFF responses confers sweep direction selectivity. The ideas are interesting but not entirely supported by the data and model in the study. Below are more specific comments.

1) The functional significance and synaptic basis of ON-OFF receptive fields in the auditory cortex has been studied in a technically challenging and detailed study by Michael Wehr in Neuron 2010 "Nonoverlapping sets of synapses drive ON responses and OFF responses in auditory cortex". Therefore, I would recommend for the authors to dial back a bit the "unknown" significance language in the abstract, and focus more on how their study complements, fits in, or differs from previous findings.

2) One key inconsistency in the study is the observation that the Hebbian model predicts that all that's required to develop non-overlapping ON-OFF RFs is exposure to sound sequences, which in the animal translates into the onset of hearing experience. However, the data from the juvenile mice (postnatal day 17-23) still show overlapping ON-OFF RFs even after 5-11 days of hearing experience (hearing onset occurs around postnatal day 12 in mice). The authors then jump to animals that are 40-70 days older (P60-90) to study RFs in adults. So the model predicts a simple switch to sound sequences drives ON-OFF RF plasticity (i.e. no structural pruning required), but the data points to a long period of over 40 days of maturation for the development of non-overlapping ON-OFF RFs... To me, this indicates that the data does not support the simple Hebbian model proposed.

- 2a) A related issue to my previous point is the false equivalency between the model's immature state and juvenile mice. In the model's immature (pre-hearing) state the authors use "noisy" inputs to simulate spontaneous cortical activity, but the juvenile mice they record from are post-hearing and are receiving patterned sound stimuli, not noise.

- 2b) Furthermore, the model is very unconstrained by Biology, and it would be very hard to reproduce with the limited information provided by the authors.

3) The authors need to explain very clearly how switching from "noisy" to "patterned" stimuli leads to the development of non-overlapping ON-OFF RFs in the frequency domain. In other words, how do you get frequency shifts from temporal offsets.

4) Although overall the study reports data from many cells, they are mainly from adult mice. There are approximately 3-times fewer neurons recorded in juvenile mice and from fewer animals, which compounds the bias. Given that the authors report modestly significant effects, these substantial differences in sample size can have a big impact on significance tests.

5) It is also not clear what fraction of all neurons recorded had OFF responses (all?). Did that fraction change between juveniles and adults?

- 5a) On a related point, in their entire population of cells (juvenile and adult) the authors should quantify and report whether there is a consistent or variable frequency bias in OFF responses. The authors mention this briefly in a small subset of neurons in the adults (Figure 4).

6) The authors don't justify reporting both CF and BF. There is no greater point made in the paper about these two metrics, and why computing both adds to the significance of their results.

7) To account for sweep direction selectivity as reported previously (Zhang et al Nature 2003): wouldn't you need a reversal in the ON-OFF RF frequency bias to account for the fact that opposite ends of the tonotopic axis have opposite sweep direction selectivity? The authors do not directly address how to account for reversals in sweep direction selectivity along the tonotopic axis.

8) The sweep responses shown in Figure 4b are of some concern. This recording has the potential to

contain multiple units. Although there are more spikes on average in UP sweeps, the "unit" is still vigorously responsive to all DOWN sweeps tested. At best this "unit" would have mixed sweep selectivity. In fact, the instantaneous firing rate does not appear very different for all sweeps tested (and it is in fact the recommended way to measure sweep responsiveness). Given the small sample size for these sweeps recordings and the potential for contamination from multiple units cast doubt on the correlation between DSI and ON-OFF responses (Figure 4d).

9) Some methodological considerations: is single unit isolation more difficult in young vs. adult cortex? In young animals cortical cells are more densely packed, their somata are smaller, and the shape of their action potentials differs from the adult. This raises the possibility of poorer single-unit discriminability.

10) The example cell shown in Figure 1d is not the most appropriate. I understand the authors chose it because it appears as a very striking example, but the ON receptive field is not fully captured on the low frequency side of the unit making the FRA incomplete.

11) The authors need to add error bars throughout many figure panels. For example: panels g, h, and i in Figure 1, and in Figure 4 for all firing rate plots.

12) The authors need to explain very clearly what is being plotted in Figure 1 panels g-i. What exactly does "Cell fraction" mean? And what are we supposed to glean from the small differences between the lines plotted. Perhaps switch the axes so that it is more intuitive.

13) The authors need to provide more information about the "non-ethological" ON-OFF simulation. This is important enough to merit being fleshed out in the Results section.

Responses to Reviewers – Sollini, Chapuis *et al.*

We are grateful to the Reviewers for their feedback and supportive remarks regarding the interest of our study.

- We have undertaken additional theoretical work to develop our model and investigate the role of cortical inhibition in the development and function of ON/OFF receptive fields.

- We have performed new pharmacogenetics experiments to test a key prediction of the model.

All together we have added **2 new Figures** and **5 Supplementary Figures** in the revised manuscript and believe this new data and analysis substantially improves the impact of the work. Detailed responses to the Reviewers comments are laid out below.

Reviewers' comments:

Reviewer #1 (Remarks to the Author):

Here Sollini et al. examine development of the mouse auditory cortex; specifically, the emergence of ON and OFF responses to pure tones. With extracellular recordings in anaesthetised animals they show that in young mice (P17-23), ON and OFF fields are more overlapping compared to adult frequency-response areas. A fairly small-scale model based on Hebbian plasticity is proposed to account for the development of ON/OFF responses, and the authors examine the relation between ON/OFF responses and FM sweep direction selectivity.

There are some interesting data here, and relating different receptive field properties, examining them over development, and trying to replicate their emergence in a modeling framework are all noble and important efforts. However, I think that this study at present is underdeveloped, suffering from a lack of connecting qualities- specifically, the authors make no attempt to test their model, which seems a bit simplistic and leaves out important aspects of processing for receptive fields (namely, inhibitory responses).

We have extended our model to include inhibitory-evoked responses, and have explicitly tested our prediction regarding the role of inhibition in direction selectivity (DS) using pharmacogenetic perturbation. Overall our results show that the arrangement of excitatory ON and OFF receptive fields are sufficient to account for DS in auditory cortex.

Critiques:

1) No effort is made to test the model experimentally. If Hebbian plasticity can fully account for the reorganization of ON/OFF receptive fields, then presenting similar sound patterns as used in the model to drive Hebbian learning (Fig. 2c) to juvenile animals, should result in more rapid divergence of ON/OFF fields in these animals. Can the authors experimentally demonstrate this? There are also predictions for spiking patterns evoked by these sounds, the potential engagement of NMDA receptors or other aspects important plasticity, but none of these issues are examined here.

We agree that it would be interesting to test several developmental aspects of the model. However, these experiments require longitudinal testing of animals that have been reared in very specific and tightly controlled acoustic environments, and we currently lack the facilities and ethical permissions to perform these experiments. We therefore respectfully suggest that such work is not feasible within the scope of our study.

The sound pattern that is used is a sequence that toggles on and off with Poisson statistics. We do not think that presenting such stimuli *in situ* will hasten the divergence of ON and OFF fields as natural acoustic environments are already characterised by abrupt onsets and offsets, but it could be possible to bias the profile of receptive field divergence by exposing animals to acoustic environments that, for example, favour onsets always closely preceding offsets (e.g. soundscapes largely composed of brief sound transients), or favour offsets closely preceding onsets (e.g. long duration sounds punctuated with brief gaps). Such scenarios may bias the stabilisation of ON versus OFF RFs (or vice versa). In the revised manuscript, we now clarify that the divergence of ON and OFF RFs is a process that occurs over multiple stimulus presentations and extended timescales (likely corresponding to days or weeks following hearing onset *in vivo*). We emphasise that changing the properties of acoustic exposure during this time is likely to subtly influence the reorganisation of RFs, but that the essential principle of ON/OFF divergence is maintained, and discuss this proposal in the new manuscript (line 282).

Our revised model does make predictions about the role of fast cortical inhibition in the divergence and functional consequences of ON and OFF RFs. We have performed new pharmacogenetic experiments to reduce fast cortical inhibition *in vivo* (using the inhibitory DREADD receptor targeted to Parvalbumin-positive, i.e. fast spiking, interneurons), and confirm the prediction that directional selectivity conferred by ON/OFF RF arrangement is preserved in this condition. These results are presented in a new Figure (**Figure 6**). We believe these experiments improve the link between theory and experiments and have substantially strengthened the manuscript.

2) ON/OFF responses have different excitatory-inhibitory relations that might shape these responses and/or depend on the maturation of inhibitory inputs, but the model takes into account only excitatory inputs. In general there's little exploration of the parameter space or important qualities of the model that might inform or constrain experiments. Inhibition might also be important for FM sweep selectivity.

We have studied the effect of inhibition by incorporating ON and OFF inhibitory inputs to our model. Inhibitory synapses were also modelled to be plastic. We used the rule of Vogels *et al.* (2011 Science), which has been validated experimentally by the Froemke group (D'Amour and Froemke, 2015). This rule has been shown to develop some amount of excitatory and inhibitory co-tuning (Vogels *et al.* 2011 Science; Clopath *et al.* BioArxiv 2016). In our model after the initial development, we observe a degree of co-tuning of excitatory and inhibitory responses, as shown experimentally by Scholl *et al.* (2010 Neuron). This match is important to make sure our model is sufficiently constrained.

Interestingly, our previous results, 1) divergence of ON and OFF RFs and 2) its correlation with sweep selectivity index, are essentially unchanged. We include this information in a fully revised **Figure 2**, a new **Figure 3**, and a new supplementary figure (**Supplementary Figure 2**). Most importantly, if we remove inhibition in this full model, our modelling results also remain unchanged. To show predictive power of our model, we decided to test by suppressing inhibition experimentally. Our new results show that experimentally too, our results do not depend on inhibition (**Figure 6**). We therefore not only show that inhibition is not the main mechanism at hand here, but also validate our model by testing our theoretical predictions.

We have also performed a parameter sensitivity analysis for the model. Specifically, we changed the learning rates, the target constant in the inhibitory plasticity rule (ρ) and the firing baseline of the cells. Systematically, we find that the divergence of ON and OFF RFs and the correlation with sweep selectivity is maintained when we change model parameters. We include this information in **Supplementary Figures 3 and 4**. Overall, this additional work

indicates that we have uncovered a very robust mechanism that does not depend on a specific implementation.

3) The baseline activity in the recording shown from the juvenile animal is much higher compared to that shown from an adult (Fig. 1a,b). Is this representative? Increased baseline activity in juvenile animals could potentially account for the ON/OFF fields overlap by imposing high-pass filtering of sensory-evoked responses. Is there a way to show that Hebbian plasticity, and not the decrease in baseline activity, is indeed responsible for the divergence of ON/OFF receptive fields in adult animals?

We have explored this issue in our experimental and modelling data. Experimentally, we observed weak but highly significant correlations between spontaneous neuronal firing rate and the amount of ON/OFF overlap within both young and adult populations (Pearson correlation coefficient, juveniles: $r = 0.2709$, $p = 0.008$, adults: $r = 0.1904$, $p < 0.001$), demonstrating that changes in baseline can modulate the amount of FRA overlap observed. However, we found no significant difference between the baseline rates in our young and adult populations (Wilcoxon rank-sum, $p = 0.8$). In our model, we systematically varied the baseline activity and found that the segregation of ON and OFF RFs is preserved. We also note that measurement of characteristic frequency, which we use most commonly to quantify ON/OFF segregation should not be sensitive to changes in baseline activity. We therefore rule out changes in baseline activity as the driver for developmental RF divergence.

4) Stereotaxic coordinates and recording depth should be included in Methods. It will be helpful to include histological verification of the recording sites in the results.

We have added this information to the Methods (line 334). We provide an example of histological verification of probe insertion here (**Review Figure 1**), but we did not label our probes with Di-I in every experiment (in part because the fluorescence could overwhelm mCherry signal in DREADD experiments). We provide histological verification of our target injection area in **Figure 6a**.

Review Figure 1: Histological verification of recording site from Adult C57BL6 mouse.

5) The 'n' is a bit low here; the critical data come from a total of four juvenile animals.

We have made recordings from additional juvenile animals. The total N for the study now stands at:

ON/OFF RF Structure

Adult C5BL/6 – N = 6

Young C5BL/6 – N = 7

ON/OFF RFs vs DS

Adult C5BL/6 – N = 4

Adult PV-Cre, transduced with FLEXed inhibitory hM4i receptor - N = 7

DREADD CONTROL

Adult C5BL/6 – N = 3

Total: 20 Adult (P60-90; 13 C57BL/6 and 7 *Pvalb*-IRES-Cre) and 7 Young (P17-23; C57BL/6)

This information is stated in the Methods (line 330).

Reviewer #2 (Remarks to the Author):

In “Hebbian plasticity underlies developmental divergence of ON-OFF receptive fields and direction selectivity in auditory cortex”, Sollini et al examine the development and functional significance of ON-OFF responses in mice. Using extracellular recordings in anesthetized juvenile and adult animals, they propose that the frequency shift between ON-OFF responses arises from hearing experience, and that a general Hebbian learning model can account for this shift. Furthermore, they propose that frequency shifts in ON-OFF responses confers sweep direction selectivity. The ideas are interesting but not entirely supported by the data and model in the study. Below are more specific comments.

1) The functional significance and synaptic basis of ON-OFF receptive fields in the auditory cortex has been studied in a technically challenging and detailed study by Michael Wehr in Neuron 2010 “Nonoverlapping sets of synapses drive ON responses and OFF responses in auditory cortex”. Therefore, I would recommend for the authors to dial back a bit the “unknown” significance language in the abstract, and focus more on how their study complements, fits in, or differs from previous findings.

We have changed the text in the abstract to try and address this point. However, we note that both developmental basis and functional significance of ON/OFF RF arrangement are unknown. Also, we are huge fans of the Scholl *et al.* paper and cite it several times. The discovery that ON and OFF responses are represented by different populations of A1 synapses is of fundamental importance to our study!

2) One key inconsistency in the study is the observation that the Hebbian model predicts that all that’s required to develop non-overlapping ON-OFF RFs is exposure to sound sequences, which in the animal translates into the onset of hearing experience. However, the data from the juvenile mice (postnatal day 17-23) still show overlapping ON-OFF RFs even after 5-11 days of hearing experience (hearing onset occurs around postnatal day 12 in mice). The authors then jump to animals that are 40-70 days older (P60-90) to study RFs in adults. So the model predicts a simple switch to sound sequences drives ON-OFF RF plasticity (i.e. no structural pruning required), but the data points to a long period of over 40 days of maturation for the development of non-overlapping ON-OFF RFs... To me, this indicates that the data does not support the simple Hebbian model proposed.

• 2a) A related issue to my previous point is the false equivalency between the model’s immature state and juvenile mice. In the model’s immature (pre-hearing) state the authors use “noisy” inputs to simulate spontaneous cortical activity, but the juvenile mice they record from are post-hearing and are receiving patterned sound stimuli, not noise.

The 'Young' state in previous version of the manuscript is indeed 'Hearing Onset', and we accept that does not correspond well with the experimental data. We have added an intermediate step in the model simulation where 'Young' is an early stage in the evolution of the population divergence (which we postulate corresponds to the first few days after Hearing Onset). We present this data in **Figure 2d**. We also add the time course of the simulation in **Figure 2f** – this data demonstrates that the divergence of ON and OFF RFs does not occur as a simple switch, but rather evolves over a prolonged period of stimulus presentation. The 'Young' neural data presented in Figure 1 was collected in the days immediately following Hearing Onset and therefore corresponds to the ON/OFF RFs on the rising part of the curve in 2f. We hope the issue of false equivalency has now been resolved.

• 2b) Furthermore, the model is very unconstrained by Biology, and it would be very hard to reproduce with the limited information provided by the authors.

We previously attempted to use the simplest possible model, with a minimal number of assumptions, to test the general nature of the learning mechanism. We have since added inhibition to our model – which is co-tuned with excitation (but not perfectly, as reported by Scholl *et al.*, 2010), and introduced the inhibitory learning rule uncovered in auditory cortex by D'Amour and Froemke (*Neuron* 2015). We find that adding inhibition does not alter the reorganisation of ON and OFF receptive fields during learning. We have added additional information about the model in the methods – clarifying where we have used experimental findings for constraint – and have deposited the model online on ModelDB with private access for Reviewers:

<http://senselab.med.yale.edu/ModelDB/showModel.cshtml?model=231851>;
Model number: 231851; Password: Reviewer123

We can make the model available to all immediately upon publication.

We believe the fidelity of our model is reflected in its predictive power: in particular (1) the arrangement of ON and OFF receptive fields confers direction selectivity, (2) fast cortical inhibition is not required for direction selectivity (**Figure 6**). Furthermore, we show the model is robust to parameterisation (included as new **Supplementary Figures 3 and 4**).

3) The authors need to explain very clearly how switching from “noisy” to “patterned” stimuli leads to the development of non-overlapping ON-OFF RFs in the frequency domain. In other words, how do you get frequency shifts from temporal offsets.

We explain this in a new section of the Discussion (line 263).

4) Although overall the study reports data from many cells, they are mainly from adult mice. There are approximately 3-times fewer neurons recorded in juvenile mice and from fewer animals, which compounds the bias. Given that the authors report modestly significant effects, these substantial differences in sample size can have a big impact on significance tests.

We tested whether differences in the sample sizes of our populations could account for the differences found in CF and FRA overlap by bootstrapping the data (random sample with replacement, 500 repeats), matching the number of cells in both populations ($n = 258$, the Young population). We performed a Kolmogorov-Smirnov test (as before) to compare our two populations for each repeat. We found a significant difference ($p < 0.01$) between young and old for CF and overlap on every single repeat tested, demonstrating that when using matched population sizes the result is maintained.

5) *It is also not clear what fraction of all neurons recorded had OFF responses (all?). Did that fraction change between juveniles and adults?*

We now include this information in the Results (line 79).

• 5a) *On a related point, in their entire population of cells (juvenile and adult) the authors should quantify and report whether there is a consistent or variable frequency bias in OFF responses. The authors mention this briefly in a small subset of neurons in the adults (Figure 4).*

There is indeed a frequency bias and we include this information in the Results (line 93) and a new **Supplementary Figure 1**. The organisation is consistent with observations made by Scholl et al. (Neuron 2010 – see Figure 2F), although their data set contains only low frequency neurons, where OFF CF is always higher than ON CF. We see a reversal at mid-range frequencies, meaning that high frequency neurons show OFF CFs lower than ON.

6) *The authors don't justify reporting both CF and BF. There is no greater point made in the paper about these two metrics, and why computing both adds to the significance of their results.*

We accept that including two measures did not add much in terms of significance, so we now only report CF for ON/OFF comparisons.

7) *To account for sweep direction selectivity as reported previously (Zhang et al Nature 2003): wouldn't you need a reversal in the ON-OFF RF frequency bias to account for the fact that opposite ends of the tonotopic axis have opposite sweep direction selectivity? The authors do not directly address how to account for reversals in sweep direction selectivity along the tonotopic axis.*

We have explored this point from several angles. Firstly, we did find ON/OFF segregation to be organised across the tonotopic axis. If we bin our neurons according to ON CF we find a clear relationship whereby OFF CF is higher than ON CF at low frequencies, and OFF CF is lower than ON CF at high frequencies (**Supplementary Figure 1**). This arrangement is observed in the subpopulation of cells where we also measure direction selectivity and is also consistent with Scholl *et al.* (Fig 2F). Interestingly, we find that ON/OFF CF arrangement and DS both move in the opposite direction to that described by Zhang et al. (see **Review Figure 2** below). This was surprising to us, but we are convinced the result is robust as it was observed by two researchers working independently, and also at two different developmental stages (see **Supplementary Figure 1**).

The key difference between our study and Zhang *et al.* is the speed at which DS is observed. Zhang *et al.* observed DS at fast, but not slow speeds (Zhang et al, 2003 *Nature* – see Fig.1c) whereas we observe DS at slow rates (**Figure 5c**; also as in Refs38-40). The inverted relationship we observe compare to the Zhang et al Nature 2003 study might arise from differences in the recording location (Tsukano et al, 2017 *Frontiers*). Alternatively, we propose that the topography (and potentially, mechanism) of slow and fast FM representation may show fundamentally different organisation in A1. ON/OFF organisation and DS at slow rates favour slow downward transitions at lower frequencies, and slow upward transitions at higher frequencies and the sweep speeds for which we observe selectivity are ethologically relevant (Grimsley et al. 2016 *Frontiers*).

Review Figure 2: DSI (measured at 2.2 oct/s) vs $\log_2(\text{CF}/7\text{kHz})$ for the 30 selected units in Fig 5, r: correlation coefficient, p: p-value.

8) The sweep responses shown in Figure 4b are of some concern. This recording has the potential to contain multiple units. Although there are more spikes on average in UP sweeps, the “unit” is still vigorously responsive to all DOWN sweeps tested. At best this “unit” would have mixed sweep selectivity. In fact, the instantaneous firing rate does not appear very different for all sweeps tested (and it is in fact the recommended way to measure sweep responsiveness). Given the small sample size for these sweeps recordings and the potential for contamination from multiple units cast doubt on the correlation between DSI and ON-OFF responses (Figure 4d).

We have looked carefully at this unit and cannot find evidence for contamination. The spontaneous and firing rate of this unit is rather low (0.39 ± 0.04 Hz), with only approximately 2 spikes per sweep for the preferred speed/direction. To convince readers of the quality of our unit isolation, we include spike waveforms and autocorrelograms for all exemplar units in **Supplementary Figure 5**. We also report the baseline firing rates for these units; these values are qualitatively low and not representative of multi-unit or contaminated activity.

We used 30 stimulus repeats for all UP and DOWN conditions, which we consider a reasonable sample number. Since the stimuli are presented in a random order, any contamination would most probably not bias the unit’s response consistently towards one particular FM direction. We also find it difficult to see how spike contamination could contribute towards the strong relationship between ON/OFF RF and DS – the expectation would be that this would degrade any relationship (e.g. by simply lowering DSI values).

We have used a simple and conservative metric to assess direction selectivity. DSI was defined as $(r_1 - r_2)/(r_1 + r_2)$, where r_1 is the mean number of spikes triggered by the UP sweep at a given FM speed, and r_2 the mean number of spikes triggered by the DOWN sweep at the same speed. The time window (w) for measuring spiking responses started at sound onset and ended 100 ms after sound offset; the time window was thus equal for UP (w_1) and DOWN (w_2) conditions, i.e. $w_1 = w_2 = w$. We chose this method to compute the DSI rather than using instantaneous firing rate, as it provides a DSI measure independent of spike timing. We acknowledge that our DSI measure differs slightly from those reported in previous studies, but this choice does not explain discrepancies with previous observations, for example the lack of selectivity we report for single units at high FM speeds.

9) Some methodological considerations: is single unit isolation more difficult in young vs. adult cortex? In young animals cortical cells are more densely packed, their somata are smaller, and the shape of their action potentials differs from the adult. This raises the possibility of poorer single-unit discriminability.

We measured unit quality using isolation distance for young and adult units with the method from Harris et al., 2001 (Neuron). Surprisingly (as we agree with the Reviewer's reasoning), we found that isolation distance was significantly higher in the juvenile population compared with the adult population (two-sample t-test, $p = 6.7898 \times 10^{-6}$, $\mu_{\text{adult}} = 19.01$, $\mu_{\text{juvenile}} = 29.8$), although the yield per animal was much lower in the young versus old animals (37 vs 109 cells per animal, respectively).

10) The example cell shown in Figure 1d is not the most appropriate. I understand the authors chose it because it appears as a very striking example, but the ON receptive field is not fully captured on the low frequency side of the unit making the FRA incomplete.

We have replaced the cell in Figure 1d with an example where the full FRA is captured.

11) The authors need to add error bars throughout many figure panels. For example: panels g, h, and i in Figure 1, and in Figure 4 for all firing rate plots.

We have done this for firing rate plots in Figures 5 and 6. The panels in Figure 1 are histograms associated with Figure 1e and don't have error bars (however we show the entire data set in Figure 1e).

12) The authors need to explain very clearly what is being plotted in Figure 1 panels g-i. What exactly does "Cell fraction" mean? And what are we supposed to glean from the small differences between the lines plotted. Perhaps switch the axes so that it is more intuitive.

We have changed the y axis labels to 'Population Fraction'. We looked into using cumulative histograms to plot this data, but think it is easier to make comparisons (e.g. between Figures 1f, 2h and 3d) if we plot the fraction of neurons that exhibit a given ON/OFF CF difference.

13) The authors need to provide more information about the "non-ethological" ON-OFF simulation. This is important enough to merit being fleshed out in the Results section.

We have fleshed this out in the Results (from line 137) and added this information as part of a new **Figure 3**.

Reviewers' comments:

Reviewer #1 (Remarks to the Author):

The authors have adequately addressed the critiques of the previous version of this manuscript. This is a solid and interesting contribution, with new data and a rare combination of experiments in young animals, together with modeling.

Reviewer #2 (Remarks to the Author):

The model in the revised version is still unconstrained for the following reasons:

1) As Reviewer #1 stated, the authors need to test the model directly with a developmental manipulation of some kind. The authors write that they do not have the facilities or approval to do this. However, the tightly-controlled acoustical conditions necessary for anesthetized recordings (included in the paper) are exactly the same as what is needed for a developmental study. The only additional equipment for a week-long rearing experiment in a sound attenuation chamber is a small tube to circulate fresh air and a light on a timer.

2) The only experimental manipulation the authors did carry out was using DREDDs. An unfortunate choice to test whether inhibition plays a role because this technique has now been shown to not work at all as advertised (Gomez et al, Science 2017: Chemogenetics revealed: DREDD occupancy and activation via converted clozapine). In other words, if inhibition plays a role it's still not settled.

It's hard to even think about what piece of data would falsify the model. And if there were, it would be easy enough to "re-adjust" some synaptic weight.

The authors do not resolve the timing issue: if only simple Hebbian mechanisms are responsible (changes in synaptic weights only), then why are RFs still overlapping up to 10 days after hearing onset (p23, the 'young' animals)? Why does it take this long?

Finally, the authors' response to my sweep direction selectivity comment (#7) is not clear (especially the x-axis of Reviewer Fig #2.) They do see that ON/OFF CFs do show tonotopic reversals, but this doesn't translate to sweep direction selectivity differences in OFF responses. To account for this observation the authors raise the provocative idea that "Alternatively, we propose that the topography (and potentially, mechanism) of slow and fast FM representation may show fundamentally different organisation in A1." This is very interesting, but if the authors do think this could be the case, then they should really pursue it to get closer to mechanisms, which are lacking in this study.

Reviewer #2 (Remarks to the Author):

The model in the revised version is still unconstrained for the following reasons:

1) As Reviewer #1 stated, the authors need to test the model directly with a developmental manipulation of some kind. The authors write that they do not have the facilities or approval to do this. However, the tightly-controlled acoustical conditions necessary for anesthetized recordings (included in the paper) are exactly the same as what is needed for a developmental study. The only additional equipment for a week-long rearing experiment in a sound attenuation chamber is a small tube to circulate fresh air and a light on a timer.

There are strong logistical and ethical reasons why the approach suggested by the reviewer is not feasible. All animal work in this study is performed under the strict conditions of a Home Office Project Licence (PPL number 70/8837). This licence does not currently permit procedures that involved prolonged alteration of acoustic environments (i.e. isolated housing in silence for several days/weeks). According to UK and EU law, our animals must be housed in comfortable conditions with regular access to food, water and fresh bedding. Our anesthetised recording chamber is not an appropriate environment to house animals for longer than 6 hours and this is an explicit legal requirement of our Project Licence. If animals were to be kept in such a place, then the box would need to be opened regularly for observation, feeding and cleaning, which would destroy the controlled acoustic environment necessary for such an experiment. It is also an institutional and UK legal requirement that animals must be kept in internally ventilated cages (IVCs) to protect animal technicians and researchers from allergen exposure. All animals are therefore housed in special rooms with appropriate ventilation - both inside individual cages, and outside in the room. This ventilation is necessarily noisy and major refurbishment, cage modification, and testing would be required to produce an appropriate acoustic environment for developmental manipulation.

Any changes to animal living environments require the scrutiny and permission of our institutional ethics committee and the UK Home Office. These permissions would not be granted without solutions to the problems outlined above and would in any case introduce additional delays while the proposal was reviewed. In short, controlled developmental manipulation is not feasible in the short or medium term, a situation appreciated by Reviewer 1 who originally raised this point.

2) The only experimental manipulation the authors did carry out was using DREDDs. An unfortunate choice to test whether inhibition plays a role because this technique has now been shown to not work at all as advertised (Gomez et al, Science 2017: Chemogenetics revealed: DREADD occupancy and activation via converted clozapine). In other words, if inhibition plays a role it's still not settled.

We respectfully disagree with the reviewer on this point. The study by Gomez *et al.* [1] provides important information regarding the mechanisms underlying the activation of DREADDs *in vivo*. Briefly, it is shown that the designer drug, clozapine-*N*-oxide (CNO) is converted to clozapine *in vivo*, and that clozapine itself can activate DREADDs. However, this means that in our hands injection of CNO will still lead to activation of hM4 inhibitory DREADD receptors, the expression of which is restricted to Parvalbumin-positive interneurons (**Figure 6a**). Accordingly, we measured strong neural activity changes following CNO injection in mice expressing DREADDs in PV interneurons. We show via several different measures (the amplitude of evoked local field potential, the spontaneous and evoked multiunit firing rates; **Figures 6b-h**) that these changes are absent when CNO was injected in wild type mice. Therefore, the effects we observe can only be attributed to DREADD activation, and not activation of endogenous receptors that might also bind clozapine. These changes in DREADD mice are consistent with a reduction in PV-mediated inhibitory drive; notably, we measured an increase in spiking activity and a sharpening of LFP deflection following CNO injection, similarly to [2,3] (using optogenetics to inactivate PV cells) and [4] (using DREADD to inactivate PV cells). As mentioned in the study by Gomez *et al.*, there is a “substantial number of publications reporting successful use of DREADDs”. We provide here a list of exemplary studies displaying successful use of the inhibitory DREADD to control the activity

of interneurons *in vivo* [4-6]. We therefore consider that pharmacogenetics is a valid technique to manipulate neural activity *in vivo*.

To summarise:

- 1) Our control data confirm that 5mg/kg CNO injection in control animals does **not** alter neural activity in A1, ruling out non-specific effects of metabolically converted CNO (i.e. clozapine acting upon other receptors in the brain) influencing our results. This information is shown in **Figures 6d and 6g**.
- 2) We provide direct immuno-histochemical evidence that expression of the hM4 receptor is **restricted** to PV+ interneurons in auditory cortex. This is shown in **Figure 6a**.
- 3) We demonstrate via three different metrics that injection of CNO in hM4-PV mice is associated with increased neural excitability in A1. These results are completely consistent with **selective inhibition of hm4-expressing PV+ interneurons** in the auditory cortex. This information is shown in **Figures 6b-h**.

Overall we have allocated considerable space in Figure 6 to demonstrate the fidelity of our experimental manipulation. We now also discuss these issues explicitly in the Results and Discussion of the revised manuscript (from lines 229 and 311).

References:

1. Gomez *et al.* (2017), Chemogenetics revealed: DREADD occupancy and activation via converted clozapine; *Science*.
2. Atallah *et al.* (2012), Parvalbumin-Expressing Interneurons Linearly Transform Cortical Responses to Visual Stimuli; *Neuron*.
3. Zhu *et al.* (2015), Control of response reliability by parvalbumin-expressing interneurons in visual cortex, *Nature Commun.*
4. Kaplan *et al.* (2016), Contrasting roles for parvalbumin-expressing inhibitory neurons in two forms of adult visual cortical plasticity; *eLife*.
5. Zou *et al.* (2016), DREADD in Parvalbumin Interneurons of the Dentate Gyrus Modulates Anxiety, Social Interaction and Memory Extinction; *Curr Mol Med*.
6. Soumier *et al.* (2014), Opposing Effects of Acute *versus* Chronic Blockade of Frontal Cortex Somatostatin-Positive Inhibitory Neurons on Behavioral Emotionality in Mice; *Neuropsychopharmacology*.

It's hard to even think about what piece of data would falsify the model. And if there were, it would be easy enough to "re-adjust" some synaptic weight.

Our model makes clear experimental predictions on the functional implications of ON/OFF field arrangement, i.e. sweep selectivity. This theoretical prediction came first and helped us design new experiments to test our model. Finally, the data supported our theory. This is a rare example where 1) we do *pre*-diction and not *post*-diction, 2) the predictions are confirmed experimentally, 3) the theory helped to design new experiments we would not have conducted otherwise. We feel this is an example of a constructive experimental/modelling collaboration.

The authors do not resolve the timing issue: if only simple Hebbian mechanisms are responsible (changes in synaptic weights only), then why are RFs still overlapping up to 10 days after hearing onset (p23, the 'young' animals)? Why does it take this long?

The developmental changes we characterise take so long because Hebbian learning is slow. Hebbian learning is not one-shot learning. Hebbian learning is based on plasticity experiments showing that a synapse needs a lot of repeated stimulation to get a significant weight change. This gives us an idea of the typical learning rate of Hebbian learning (for more details, see fitting of learning rate and parameters in Clopath *et al.*, *Nat. Neurosci*

2010). The learning rate is such that it needs days until we can see a difference in ON and OFF separation. Finally, note that the separation undergoes a type of symmetry breaking which means that it takes a lot of time to see the beginning of the effect, and then once the symmetry is broken, the changes are more rapid (see Kempter *et al.* Physical Review E 1999 for analytical results). We have added this information to the Discussion (from line 286).

Finally, the authors' response to my sweep direction selectivity comment (#7) is not clear (especially the x-axis of Reviewer Fig #2.) They do see that ON/OFF CFs do show tonotopic reversals, but this doesn't translate to sweep direction selectivity differences in OFF responses. To account for this observation the authors raise the provocative idea that "Alternatively, we propose that the topography (and potentially, mechanism) of slow and fast FM representation may show fundamentally different organisation in A1." This is very interesting, but if the authors do think this could be the case, then they should really pursue it to get closer to mechanisms, which are lacking in this study.

To clarify our response to comment 7, we do see that ON/OFF RF arrangement and DSI are tonotopically organised (see **Supplementary Figure 1**). In the Reviewer Figure 2, the x-axis represents the octave difference between the measured CF (taken from the ON/OFF FRA – left/right plot respectively) and the lowest frequency presented (7kHz), i.e. $\log_2(\text{CF}) - \log_2(7\text{kHz}) = \log_2(\text{CF}/7\text{kHz})$, as written in the figure caption. The x-axis value of 0 represents $\text{CF} = 7\text{kHz}$, a value of 1 represents $\text{CF} = 14\text{kHz}$. Our point was that the direction of this arrangement is the opposite to that described in the study of Zhang *et al.* 2003. We have tried to emphasise that these two results do not necessary contradict each other as our study concerns units with slow speed selectivity (i.e. below 8.8 oct/sec) whereas the Zhang *et al.* study covers units with fast speed selectivity (i.e. above 16 oct/sec). We agree with Reviewer 2 that the broader issues are regarding fast and slow frequency modulation are interesting, but feel that they extend beyond the scope of the current study. We already do provide important and new mechanistic information here: specifically, we show (1) how Hebbian plasticity shapes the development of ON and OFF receptive fields, and (2) that the subsequent arrangement of ON and OFF fields confers directional selectivity in the absence of fast peri-somatic synaptic inhibition.

Reviewers' comments:

Reviewer #3 (Remarks to the Author):

Review of: Hebbian plasticity underlies developmental divergence of ON-OFF receptive fields and direction selectivity in auditory cortex by Sollini et al.

I enjoyed reading this manuscript, which presents a series of findings exploring the function of the divergence of On and Off frequency selective responses of neurons in the auditory cortex with development. Changes in auditory processing with development are understudied and I applaud the authors for tackling this topic. Initially, the authors present data to demonstrate that the On and the Off receptive fields of A1 neurons are overlapping in young animals, but become adjacent in adults. They then build a model based on Hebbian plasticity, which demonstrates that with Hebbian learning, On and Off inputs become weighted differentially with sequential On-Off-On-Off activation. Next, they propose that the difference in On and Off frequency tuning might support selectivity for frequency modulated sweeps of specific direction. Indeed, in adult mice, they find a correlation between the difference in On and Off tuning and direction selectivity index. They then add an experiment that is meant to test an alternate hypothesis that a specific type of cortical inhibitory neuron might support FM selectivity. A number of these findings are compelling, including the correspondence between On and Off frequency tuning and direction selectivity and the change in On and Off frequency tuning with development.

However, several aspects of the study appear incomplete, and some experiments seem disjointed. In addition, the titles of the figures (and the manuscript) appear to make unsubstantiated claims and the language needs to be carefully revised to capture the statistical results. For instance, I am not sure that the claim reflected in the title "Hebbian plasticity underlies developmental divergence of ON-OFF receptive fields and direction selectivity in auditory cortex" is supported by the data. I detail my concerns in order of the figures below.

1. It is unclear how neurons in young mice respond to FM sweeps. In order to support the claim that the On-Off divergence facilitates FM sweep direction selectivity, the authors should show that in young mice A1 neurons do not exhibit FM sweep direction selectivity.
2. The discussion of the test of the model in Figures 2 and 3 is somewhat confusing. First, it was unclear to me from reading the manuscript how sound stimulation was presented in the model. Only by opening the Matlab code (thank you for providing the link!) did I figure out that a random subset of channels was turned On and then Off for each event. It would be more informative if in the diagram, the authors could display the timecourse for separate input channels. Second, it is not clear to me why turning one channel On and Off independent of another channel is the less natural stimulation profile – often, there are multiple sound sources in our environment, for example, two people talking (or two mice vocalizing), which does not always happen in alternation. Also, while it's interesting that Hebbian learning can accomplish this task, are there alternative models and how does their performance rank relative to Hebbian learning?
3. The title of Figure 2 "Hebbian plasticity drives developmental divergence..." is misleading, since the authors do not actually test in vivo whether neurons exhibit plasticity, or whether suppressing Hebbian plasticity precludes divergence in development. Rather they propose a model, which is consistent with the divergence that is observed developmentally.
4. In Figure 5, the correspondence between the On/Off receptive field divergence and DSI is very compelling, however this difference accounts for only a fraction of the variance in DSI. Rather than

simply computing a correlation, would it be possible to build a model that would predict DSI based on the On and Off receptive field divergence, taking into account the distinct variability in neuronal responses to tones and FM sweeps in different cells? Can other aspects of frequency tuning, such as receptive field overlap also contribute? The analysis can be expanded further.

5. Figure 5: this analysis should be performed for data collected from young mice to test the expectation that direction selectivity should not be exhibited by neurons in which On and Off receptive fields overlap.

6. The DREADD experiment presented in figure 6 seems incomplete. If I understand correctly the motivation for the experiment, it's that it tests the finding in Figure 3CD that removing inhibition from the model does not preclude On/Off receptive field divergence. By removing inhibition with DREADDs, the authors test whether the divergence in receptive fields persists.

6.1 It seems that the time course of the DREADD inactivation is inconsistent with this goal. If I understand the motivation for the Hebbian plasticity model, it is that this plasticity drives developmental changes over weeks. However, in this experiment, DREADDs are activated 20 minutes prior to the recording – a time line inconsistent with development. Therefore, I am not convinced that this experiment would support the model findings.

6.2. It seems that this perturbation experiment does not test the center aspect of the model. It would be more powerful here to perform a perturbation experiment that would alter On and Off receptive field divergence throughout development, and demonstrate that that would also affect DSI.

6.3. The statistical quantification of the results leaves unresolved questions. The effects of suppressing PVs on On and Off receptive fields of A1 neurons should be quantified more extensively across the frequency profile – a single neuron example in Fig. 6H is insufficient (and seems to diverge from results published using optogenetic approaches to suppress PVs).

6.4 In I and J, direction selectivity should be compared in each unit before and after CNO administration as in 6H. Statistics over the population should be presented.

6.5 Why is only a single type of inhibitory interneuron tested? A number of other neurons, such as SSTs and VIPs have been shown to affect sound encoding in the auditory cortex.

7. There are a number of statements in the text unsubstantiated by the data. For example: line 67: "Specifically, ON/OFF RF arrangement governs sensitivity to slow, ethologically relevant, frequency modulations":

"governs" would imply a causal effect, which was neither tested nor shown.

Line 69: "providing a novel mechanism for cortical encoding of vocalizations"

"Potentially providing" would be more appropriate. Also it would be interesting to actually test experimentally whether indeed there is a correlation between On/Off divergence and selectivity for vocalizations.

8. EPSP should be defined.

Reviewer #3 (Remarks to the Author):

We thank the reviewer for their supportive and constructive remarks. We have performed new experiments (recording responses to FM sweeps in young mice), modelling (testing alternative schemes of input and plasticity rules), and analysis (relating direction selectivity to other neuronal properties). We have also carefully revised the manuscript to ensure that all claims are accurate with respect to the experimental and modelling data.

1. It is unclear how neurons in young mice respond to FM sweeps. In order to support the claim that the On-Off divergence facilitates FM sweep direction selectivity, the authors should show that in young mice A1 neurons do not exhibit FM sweep direction selectivity.

We have now tested FM sweep selectivity in young mice and include this information in **Figure 5D**. We find that there is some directional selectivity in young animals but the distribution of DSI values, and their relationship to ON/OFF RF organisation differs substantially from adult. We observe a bias towards upward sweep selectivity, (which was also reported in a previous study; Carrasco et al. 2013, and as expected in young animals, the difference between ON and OFF CF is a poor predictor of DS (see response to points 4 and 5 below). We interpret these data as indicating that subcortical mechanisms of FM direction selectivity are active immediately at hearing onset, while ON-OFF RF divergence contributes an additional mechanism of direction selectivity during developmental maturation of the cortex. As recommended, we have changed the language throughout the manuscript to clarify this point.

2. The discussion of the test of the model in Figures 2 and 3 is somewhat confusing. First, it was unclear to me from reading the manuscript how sound stimulation was presented in the model. Only by opening the Matlab code (thank you for providing the link!) did I figure out that a random subset of channels was turned On and then Off for each event. It would be more informative if in the diagram, the authors could display the timecourse for separate input channels. Second, it is not clear to me why turning one channel On and Off independent of another channel is the less natural stimulation profile – often, there are multiple sound sources in our environment, for example, two people talking (or two mice vocalizing), which does not always happen in alternation.

We have produced a revised version of the model where multiple sound sources (represented by independent activation within different frequency channels) are presented simultaneously. Importantly we find that this pattern of stimulation does not change any of the results, and divergence of ON and OFF CF still takes place. We have modified **Figure 2B** to add the time course for separate input channels in this stimulation scheme, and included the summary data for the ‘Overlapping sound input’ scheme in **Supplementary Figure 3a**. We include information on both stimulation patterns in the Results and Methods and include both versions model online.

Also, while it's interesting that Hebbian learning can accomplish this task, are there alternative models and how does their performance rank relative to Hebbian learning?

We explored an alternative plasticity rule that is present in cortex, namely homeostatic synaptic scaling. In this scenario, ON and OFF RFs do not diverge (see **Supplementary Figure 4**).

3. The title of Figure 2 “Hebbian plasticity drives developmental divergence...” is misleading, since the authors do not actually test in vivo whether neurons exhibit plasticity, or whether suppressing Hebbian plasticity precludes divergence in development. Rather they propose a model, which is consistent with the divergence that is observed developmentally.

We have changed the titles of the manuscript and Figure 2 to read, ‘Hebbian plasticity can *underlie/account for* developmental divergence...’ to reflect this point.

4. In Figure 5, the correspondence between the On/Off receptive field divergence and DSI is very compelling, however this difference accounts for only a fraction of the variance in DSI. Rather than simply computing a correlation, would it be possible to build a model that would predict DSI based on the On and Off receptive field divergence, taking into account the distinct variability in neuronal responses to tones and FM sweeps in different cells? Can other aspects of frequency tuning, such as receptive field overlap also contribute? The analysis can be expanded further.

To test the contribution of different neural properties in predicting the DSI, we generated linear multi-variable models based on the unit properties recorded in adult mice. The importance of each property in predicting the DSI was assessed via two parameters, (1) the absolute value of the normalised coefficient, and (2) the proportional reduction of error (PRE) generated when adding the property as predictor into the model. The performance of the model in predicting the DSI was assessed via the adjusted R-square, which takes into account the number of parameters used.

First, we built a model encompassing all neural properties judged relevant (n=10 properties, namely the firing rate of the ON response to tones of CF_{on} frequency at 60dB, the firing rate of the OFF response to tones of CF_{off} frequency at 60dB, the Fano factor of each of these two responses, the spontaneous firing rate and associated Fano factor, the bandwidths of ON and OFF RFs (measured at 30dB), the percentage overlap between ON and OFF RFs, and the octave difference between ON and OFF CFs). This model was called the **Full** model. The adjusted R-square of the model was 0.279, which was worse than the adjusted R-square of the single-variable model built using the ON-OFF CF difference as sole predictor (adj. R-square=0.348). Note that in the Full model, the difference between ON and OFF CF has the largest coefficient value (see Table 1), and that it accounted for a large portion of the error reduction.

Prop.	FRon	FRoff	FFon	FFoff	FRsp	FFsp	BWon	BWoff	overlap	diff CF
Coeff.	-0.021	0.007	-0.012	0.001	0.025	0.002	-0.090	0.151	-0.194	-0.355
PRE	1.257	0.100	0.365	0.002	0.100	0.009	0.733	2.536	0.774	0.907

Table 1: Normalised coefficients and proportional reduction of error for the **Full** model. PRE for variables considered highly predictive are marked in yellow. Adj. R-square 0.279.

The lower R-square in the Full model could originate from spurious variables which do not add any predictive information. We thus removed the variables considered to have a low PRE (see Table 1), and generated a new model, called the **Partial** model. Once again, the difference between ON and OFF CF had the strongest normalised coefficient (see Table 2), and accounted for a large portion of the error reduction. The adjusted R-square of this model improved over the Full model (adj. R-square=0.400).

Prop.	FRon	FRoff	FFon	FFoff	FRsp	FFsp	BWon	BWoff	overlap	diff CF
Coeff.	-0.012	/	/	/	/	/	-0.033	0.131	-0.168	-0.330
PRE	0.409	/	/	/	/	/	0.087	1.769	0.516	0.731

Table 2: Normalised coefficients and proportional reduction of error for the **Partial** model. PRE for variables considered highly predictive are marked in yellow. Adj. R-square 0.400.

By iteratively adding and removing variables from the Partial model, the **Optimal** model was found to be composed of the following predictors: the firing rate of the ON response to tones of CF_{on} frequency at 60dB, the bandwidth of the OFF RF, the percentage overlap between ON and OFF RFs, and the octave difference between ON and OFF CFs. The coefficients and PRE for the predictors of this model are presented in Table 3. This model had an adjusted R-square of 0.419. Any addition or removal of predictor to this model decreased the adjusted R-square.

Prop.	FRon	FRoff	FFon	FFoff	FRsp	FFsp	BWon	BWoff	overlap	diff CF
Coeff.	-0.012	/	/	/	/	/	/	0.123	-0.169	-0.323
PRE	0.380	/	/	/	/	/	/	1.526	0.517	0.691

Table 3: Normalised coefficients and proportional reduction of error for the **Optimal** model. PRE for variables considered highly predictive are marked in yellow. Adj. R-square 0.419.

Whilst the Optimal model outperformed the single-variable model using the difference between ON and OFF CFs as sole predictor, it should be noted that the increase in adjusted R-square was rather small (0.07). Also, these extra predictors (the firing rate of the ON response to tones of CF_{on} frequency at 60dB, the bandwidth of the OFF RF, the percentage overlap between ON and OFF RFs) performed poorly when used as single-variable predictors (adjusted R-square -0.006, 0.046 and -0.018 for each variable respectively).

We therefore conclude that the difference between ON and OFF CFs is the best linear predictor for the DSI, but that other neural properties (presented as part of the Optimal model) can account for the DSI variability to a minor extent. This information has been added to the manuscript, and the model information is included in **Supplementary Tables 1 & 2**.

5. Figure 5: this analysis should be performed for data collected from young mice to test the expectation that direction selectivity should not be exhibited by neurons in which On and Off receptive fields overlap.

To predict the DSI of the cells recorded in the young mice, we used the linear models built based on the adult cells properties (see Table 3). We used both the Optimal model, and the single-variable model based on the difference between ON and OFF CFs as single predictor. We found that for both Optimal and single-variable models, the distributions of DSI were centered around 0 (median \pm m.a.d. of DSI distributions: Optimal: -0.047 ± 0.10 ; single-

variable: -0.010 ± 0.08). Our prediction was thus consistent with direction selectivity being weakly exhibited by neurons with low ON-OFF CF difference.

To further show that the properties found in Young neurons could not account for the DSI, we developed a Full model (using the same parameters as for Adult neurons). In the case of Young neurons, the difference between ON-OFF CFs was not a good predictor of the DSI (see Table 4). Instead, the percentage overlap was found to be the variable with the best PRE. However, the percentage overlap was not a good predictor of DSI when used as single-variable model (adjusted R-square 0.059) indicating that there was no clear linear relationship between DSI and any of the Young neuron properties. This information has been added to the manuscript, and the model information is included in **Supplementary Table 3**.

Prop.	FRon	FRoff	FFon	FFoff	FRsp	FFsp	BWon	BWoff	overlap	diff CF
Coeff.	-0.004	0.008	0.01	-0.028	0.207	0.025	-0.120	0.066	-0.638	-0.151
PRE	0.008	0.015	0.055	0.956	0.493	1.080	1.111	0.173	4.370	0.077

Table 4: Normalised coefficients and proportional reduction of error for the **Full** model (Young). PRE for variables considered highly predictive are marked in yellow. Adj. R-square -0.339.

6. The DREADD experiment presented in figure 6 seems incomplete. If I understand correctly the motivation for the experiment, it's that it tests the finding in Figure 3CD that removing inhibition from the model does not preclude On/Off receptive field divergence. By removing inhibition with DREADDs, the authors test whether the divergence in receptive fields persists.

6.1 It seems that the time course of the DREADD inactivation is inconsistent with this goal. If I understand the motivation for the Hebbian plasticity model, it is that this plasticity drives developmental changes over weeks. However, in this experiment, DREADDs are activated 20 minutes prior to the recording – a time line inconsistent with development. Therefore, I am not convinced that this experiment would support the model findings.

The aim of this experiment (both the model and *in vivo*) was to test whether intact synaptic inhibition is required for direction selectivity in the cortex. The prevailing hypothesis since Zhang, Tan *et al.* Nature 2003 has been that DS responses in A1 result from asymmetrical tuning of cortical inhibition (w.r.t. sound frequency). Therefore, frequency sweeps entering the excitatory region of the neuron's FRA will activate fast synaptic inhibition prior to excitation in one direction, quenching the excitatory response for either upward or downward FM. Our model shows that asymmetrically tuned inhibition (so called 'side band inhibition') is not necessary for slow DS in A1. We tested this experimentally by transiently perturbing synaptic inhibition from one class of inhibitory interneuron, predicting that DS would be preserved during the perturbation. Our results demonstrate that the arrangement of ON and OFF RFs, in the absence of fast perisomatic inhibition, is still a good predictor of DS, supporting our modelling data. We do not wish to make any claims about the role of inhibition during development, only that inhibition mediated by parvalbumin-positive is not required for slow DS in adult A1 – a conclusion that supports the model findings.

6.2. It seems that this perturbation experiment does not test the center aspect of the model. It would be more powerful here to perform a perturbation experiment that would alter On and Off receptive field divergence throughout development, and demonstrate that that would also affect DSI.

We agree that this would be a very interesting approach. Unfortunately these experiments are extremely difficult to perform as they require animal rearing in perfectly controlled acoustic environments for prolonged periods of time (several weeks) and/or sustained perturbation of cortical activity via pharmacological/pharmacogenetic means. In the former case, the development and testing of the acoustic environment is non-trivial and in the latter case, such perturbations are very likely to generate confounding and compensatory changes within the cortex. Both approaches also require additional ethical permission to be sought from UK government. We therefore suggest that such procedures are appropriate for future studies.

6.3. The statistical quantification of the results leaves unresolved questions. The effects of suppressing PVs on On and Off receptive fields of A1 neurons should be quantified more extensively across the frequency profile – a single neuron example in Fig. 6H is insufficient (and seems to diverge from results published using optogenetic approaches to suppress PVs).

To quantify the effect of PV suppression on the frequency profile, we measured the bandwidths of ON and OFF RFs as well as the threshold level at BF for the two populations of cells recorded pre- and post- CNO. No statistical difference was found between the distributions pre- and post- CNO ($p > 0.1$ for all variables tested, Wilcoxon rank sum test). This information has been added to the manuscript (**line 276**).

6.4 In I and J, direction selectivity should be compared in each unit before and after CNO administration as in 6H. Statistics over the population should be presented.

We attempted to compare individual units before and after CNO injection, but we confronted two important confounds. First, injection of CNO was often associated with transient electrode drift, and second, DREADD-mediated reduction of cortical inhibition unmasked many previously silent neurons in our recordings. Both factors meant that it was necessary to spike-sort pre- and post-CNO epochs separately because both the number and identity of units changed substantially. It was therefore not possible to systematically and unambiguously classify pre- and post-CNO clusters as the same unit. For this reason, we have performed population-level analysis of units before and after administration. We provide these population statistics for frequency response properties (point 6.3), along with measures of LFP and MU activity to confirm the fidelity of our perturbation.

6.5 Why is only a single type of inhibitory interneuron tested? A number of other neurons, such as SSTs and VIPs have been shown to affect sound encoding in the auditory cortex.

We tested PV inhibition on the basis of the Zhang, Tan *et al.* Nature 2003, which identified fast feedforward inhibition as a prime candidate to mediate DS in the cortex. We agree that it will be important to investigate the role of other interneuron classes in this context. However, we do not currently have access or financial resources access the appropriate transgenic mouse lines at this stage. We therefore respectfully request that these experiments are appropriate for a future study. We have added text and two references to the discussion in order to highlight the importance of this point (**line 351**).

7. There are a number of statements in the text unsubstantiated by the data. For example: line 67: "Specifically, ON/OFF RF arrangement governs sensitivity to slow, ethologically relevant, frequency modulations": "governs" would imply a causal effect, which was neither tested nor shown.

We have changed the text here and throughout the manuscript to ensure that all claims are substantiated.

*Line 69: "providing a novel mechanism for cortical encoding of vocalizations"
"Potentially providing" would be more appropriate. Also it would be interesting to actually test experimentally whether indeed there is a correlation between On/Off divergence and selectivity for vocalizations.*

Done.

8. EPSP should be defined.

Done.

REVIEWERS' COMMENTS:

Reviewer #3 (Remarks to the Author):

My concerns have been addressed. The paper is much approved.